



**A statistical examination of the effects of stratospheric sulphate geoengineering on tropical storm genesis**

**Qin Wang[1], John C. Moore[1,2,3*], Duoying Ji[1]**

[1] *College of Global Change and Earth System Science, Beijing Normal University, 19 Xinjiekou Wai St., Beijing, 100875, China*

[2] *Arctic Centre, University of Lapland, P.O. Box 122, 96101 Rovaniemi, Finland*

[3] *CAS Center for Excellence in Tibetan Plateau Earth Sciences, Beijing 100101, China*

**\* *Correspondence to*: John C. Moore. College of Global Change and Earth System Science Beijing Normal University 19 Xinjiekou Wai street, Beijing,China,100875 fax:+86-1058802165 Mobile +86-13521460942 E-mail: john.moore.bnu@gmail.com.**



**Abstract**
The thermodynamics of the ocean and atmosphere partly determine variability in
tropical cyclone (TC) number and intensity and are readily accessible from climate
model output, but a complete description of TC variability requires much more
dynamical data than climate models can provide at present. Genesis potential index
(GPI) and ventilation index (VI) are combinations of potential intensity, vertical wind
shear, relative humidity, midlevel entropy deficit, and absolute vorticity that can
quantify both thermodynamic and dynamic forcing of TC activity under different
climate states. Here we use six CMIP5 models that have run the RCP4.5 experiment
and the Geoengineering Model Intercomparison Project (GeoMIP) stratospheric
aerosol injection G4 experiment, to calculate the two TC indices over the 2020 to 2069
period across the 6 ocean basins that generate tropical cyclones. Globally, GPI under
G4 is lower than under RCP4.5, though both have a slight increasing trend. Spatial
patterns in the effectiveness of geoengineering show reductions in TC in the North
Atlantic basin, and Northern Indian Ocean in all models except NorESM1-M. In the
North Pacific, most models also show relative reductions under G4. Most models
project potential intensity and relative humidity to be the dominant variables affecting
genesis potential. Changes in vertical wind shear are significant, but both it and vorticity
exhibit relatively small changes with large variation across both models and ocean
basins. We find that tropopause temperature is not a useful addition to sea surface
temperature in projecting TC genesis, despite radiative heating of the stratosphere due
to the aerosol injection, and heating of the upper troposphere affecting static stability



and potential intensity. Thus, simplified statistical methods that quantify the
thermodynamic state of the major genesis basins may reasonably be used to examine
stratospheric aerosol geoengineering impacts on TC activity.
Key word: tropical cyclone, hurricanes, ENSO, statistical methods

**1 Introduction**
Anthropogenic greenhouse gases emission are changing climate (IPCC, 2007). The best
solution for limiting climate change is to reverse the growth in net greenhouse gases
emissions. It is doubtful that reductions in emission can be done fast enough to limit
global mean temperatures rises to targets such as the 1.5° or 2°C pledged at the Paris
climate meeting (Rogelj et al., 2015). Geoengineering is the deliberate and large-scale
intervention of Earth's climate system to retard climate warming (Crutzen, 2006;
Wigley, 2006). Geoengineering by solar radiation management (SRM) attempts to
lessen the incoming sunlight to counteract the effect of global warming. The
Geoengineering Model Intercomparison Project (GeoMIP) (Kravitz, et al.,2011) is a
standardized set of experiments designed to facilitate earth system model (ESM)
simulations of geoengineered climates, and is supported by about 12 model groups
globally, with further experiments planned under CMIP6 (Kravitz et al., 2015). Climate
system thermodynamics will certainly change under SRM geoengineering where the
reduction in short wave radiation is designed to offset increases in long wave absorption
(Huneeus et al., 2014).



Tropical cyclones (TCs) are one of the most disastrous weather phenomena
influencing agriculture, human life, and property (Chan et al., 2005). The large-scale
changes in surface temperatures under greenhouse gas forcing will impact cyclogenesis
changing both the frequency and intensity of tropical cyclones (Grinsted et al., 2012;
2013). Hence, how tropical cyclones would change in a geoengineered world is of
general as well as scientific interest for its enormous social and economic impact.
However, since almost all climate models do not, at present, possess the resolution
required to simulate directly the response of tropical cyclones to changing patterns of
radiative forcing, methods that rely on the statistical links between the thermodynamics
of the ocean and atmosphere with cyclone dynamics have been the topic of studies.

Many methods have used to study the changes in typhoons under climate warming.

Some focus on the movement of tropical storm tracks, tropical cyclone intensity and
frequency by downscaling (Emanuel, 2006). The most direct way is to use historical
climate and storm records to quantitatively study tropical cyclone activity and its
relation to key variables such as local, tropical and global sea surface temperatures, and
various teleconnection patterns (Grinsted et al., 2012; Emanuel, 2008; Landsea, 2005;
Gray, 1979). Potential intensity theory (Bister et al., 1998; Emanuel et al., 2004)
predicts the dependence of typhoon wind speed on the air-sea thermodynamic
imbalance and the temperature of the lower stratosphere. For example, many studies
suggest that wind shear has inhibitory effect on the TC activity (Vecchi et al., 2007).
Others have also identified changes in the large-scale environmental factors influencing



the tropical storm activity to assess the TC activities in the future (Tippett et al., 2011;
Grinsted et al., 2013).

While much is known about which factors influence genesis, a quantitative theory

is lacking, so empirical methods have been used to define the relationship between
large-scale environmental factors and tropical cyclogenesis. The GPI uses four
environmental variables: potential intensity, low-level absolute vorticity, vertical wind
shear, and relative humidity. Tang et al. (2012) introduced the VI, defined as the flux of
low-entropy air into a tropical disturbance or TC, because ventilation disrupts the
formation of a deep, moist column that is hypothesized to be necessary for the spin up
of the vortex (Bister et al., 1997; Nolan, 2007; Rappin et al., 2010). For the Atlantic
hurricane region, Tippett et al. (2011) formulated a genesis potential index using the
relative sea surface temperature, defined as the tropical Atlantic sea surface
temperatures minus the tropical mean sea surface temperatures, and midlevel relative
humidity in lieu of the potential intensity and non-dimensional entropy deficit,
respectively. Dynamic potential intensity (DPI) is yet another index designed to
describe the ocean's impact on tropical cyclones (Balaguru et al., 2015). These indices
represent the climatological thermodynamic spatial and seasonal control of TC genesis
and not the dynamic development of individual storms, which is more or less beyond
the abilities of contemporary climate models. The relative contribution of the individual
large-scale environmental factors to TC genesis may be different in different ocean
basins (Wang et al., 2012).



An increase in future global TC frequency has been projected based on dynamical
downscaling CMIP5 models (Emanuel, 2013). However, the same downscaling applied
to the CMIP3 models projected a decrease in global TC frequency (Tory et al., 2013;
Emanuel et al., 2006). Some models show that although Atlantic TC frequency will
decrease, the frequency of severe TC will increase, and different TC basins are
predicted to behave differently (Emanuel et al., 2008; Thomas et al., 2015; Kang et al.,

2012).

There has been little research about TC changes under geoengineering. Moore et al.
(2015) used statistical relation between Atlantic tropical storm surges and spatial
patterns of global surface temperature to deduce that moderate amounts of SRM could
reduce the frequency of the most intense hurricanes relative to greenhouse gas only
climates. Jones et al. (2017) show that applying aerosol injection to northern and
southern hemispheres separately reduced the numbers of TC in North Atlantic if the
northern hemisphere was cooled, while increasing them if aerosol was released only in
the southern hemisphere, relative to both greenhouse gas forcing both with, and without,
global stratospheric aerosol injection.
Here we examine ESM simulations of global TC evolution under stratospheric
sulphate injection geoengineering and greenhouse gas forcing based on the
climatological GPI and VI. We explore the effects of geoengineering on TC
thermodynamics, and study regional characteristics of typhoon and hurricane
development after implementation of geoengineering.



Section 2 introduces the methods and data used in this study. Section 3 describes
the temporal and spatial variations of the GPI and ventilation index in six models, in
greenhouse gas and SRM simulations. We quantify the contribution of each variable to
TC genesis using two statistical methods. Finally we study the effect of ENSO on TC
and TC track of HadGEM2-ES model. A discussion and conclusions are provided in
section 4.
**2 Methods and data**
**a.Methods**
We use climate model output from the GeoMIP G4 experiment (Kravitz et al.,
2011) and the control simulation, RCP4.5 experiment of CMIP5 (Taylor et al., 2012) to
analysis the characteristic of TC changes in the future in different models. G4 is based
on the greenhouse gas emissions from the RCP4.5 scenario but short wave radiative
forcing is reduced by injection of $SO_2$ into the equatorial lower stratosphere at a rate of
5 Tg per year from the year 2020 to 2069. The experiment continues for a further 20
years to 2089 with only greenhouse gas forcing as specified by RCP4.5. The general
climate response to G4 forcing has been discussed by Yu et al. (2015). Between 2050
and 2069, global surface air temperatures warm by 1.3 ℃ in RCP4.5, and by 0.79 ℃
with G4 relative to 2010–2029. Over the same interval, tropical North Atlantic
temperatures in the so-called Main Development Region (MDR) of cyclogenesis in the





basin warm by 0.8 ℃ and 0.4 ℃ with RCP4.5, and G4, respectively (Moore et al.,

2015).

We assess the large-scale environmental conditions for TC generation primarily in
reference to the widely used genesis potential and ventilation index (GPI), and use
results for the VI for comparison. While other indices also exist as mentioned above,
the data fields required to calculate them are presently not all available. The signal to
noise ratio of the G4 experiment is not as large as that of G1 (Yu et al., 2015) where
solar dimming offsets quadrupled $CO_2$ concentrations. It is, however, more interesting
for TC studies because the sulphate aerosol injected into the stratosphere causes
radiative heating that will potentially affect the deep tropospheric convention systems
that characterize intense tropical storms.

The GPI has been widely employed to represent TC activities (e.g., Song et al.,

2015). We use the Emanuel et al., (2004) method to calculate the GPI as follows:
$$GPI = \left|10^5 \eta\right|^{3/2} \left(\frac{H}{50}\right)^3 \left(\frac{V_{pot}}{70}\right)^3 \left(1 + 0.1V_{shear}\right)^{-2} \qquad (1)$$

where $\eta$ is the absolute vorticity in s⁻¹, $H$ is the relative humidity at 700 hPa in
percent, $V_{pot}$ is the Potential intensity in ms⁻¹, and $V_{shear}$ is the magnitude of the vector
shear from 850 to 200 hPa, in ms⁻¹. Potential intensity (Emanuel, 2000) is defined as
$$V_{pot}^2 = C_p \left(T_S - T_O\right) \frac{T_S}{T_O} \frac{C_K}{C_D} \left(\ln \theta_e^* - \ln \theta_e\right) \qquad (2)$$



Where $T_S$ is the ocean surface temperature, $T_O$ is the mean outflow temperature,
which is taken near the tropopause at the 100 hPa level and spatially averaged, $C_K$ is
the exchange coefficient for enthalpy, and $C_D$ is the drag coefficient. $\theta_e^*$ is the
saturation equivalent potential temperature at the ocean surface, and $\theta_e$ is the boundary
layer equivalent potential temperature.
We also use a second and more recent method to estimate TC called the
ventilation index (Tang, et al., 2014), defined as:
$$VI = \frac{\chi_m V_{shea}}{V_{pot}} \qquad (3)$$
Where $\chi_m$ is the (nondimensional) entropy deficit, defined as:
$$\chi_m = \frac{s_m^* - s_m}{s_{SST}^* - s_b} \qquad (4)$$
where $s_m^*$ is the saturation entropy at 600 hPa in the inner core of the TC, $s_m$ is the
environmental entropy at 600 hPa , $s_{SST}^*$ is the saturation entropy at the sea surface
temperature, and $s_b$ is the entropy of the boundary layer, which we chose as the 925
hPa layer. The numerator of (4) is the difference in entropy between the TC and the
environment at mid-levels, while the denominator is the air-sea disequilibrium, both are
calculated following Emanuel (1994).
**b.   Data**
Although to date 8 ESM have performed the greenhouse gas and G4 simulations,
we selected a subset of 6 models to use here based on access to all required model data



fields (Table 1). We use monthly sea surface temperature (SST), relative humidity,
vertical wind shear, sea level pressure, specific humidity, air temperature. All the model
outputs at different spatial resolutions were interpolated to a common grid ($128\times64$)
using the bilinear interpolation method. All the models were weighted equally in the
ensemble mean, so the models with more than a single ensemble member were first
averaged before taking the overall model ensemble mean.
**c.   TC basins**

Factors influencing TC change are diverse across different ocean basins. Some

researchers (Emanuel, 2010; Knutson et al., 2015) find a decline in the frequency of
events in the Southern Hemisphere, but increasing frequency in the Northern
Hemisphere. We therefore examine relationships across all the six TC basins listed in
Table 2. The observed TC annual mean numbers for the period 1980-2008 for each
basin (Emanuel, 2010) are also listed in Table 2. The North Atlantic makes up a
relatively small fraction of the total, with the Pacific dominant in the global locations
of tropical cyclones.
**3   Results**
**3.1 The temporal and spatial distribution of GPI and VI**

The time series of annual GPI over the 6 TC basins and during the appropriate TC

season (The Northern Hemisphere peak TC season is defined to be August through
October, and the Southern Hemisphere season is defined to be January through March.)



are shown in Fig. 1. Hereafter, all analyses are calculated and compared using these
monthly periods. The mean differences in the TC indices and their component parts are
tabulated in Table 3.
The GPI has a rising trend, significant at the 95% level, for all models except BNU-
ESM and CanESM2 under RCP4.5, and for all models except CanESM2 and
NorESM1-M under G4. Furthermore, the G4 means for all models were significantly
lower than their RCP4.5 values. The models we use have considerable range in their
absolute values of GPI, which is also a generally observed feature of climate models
(Emanuel, 2013). The MIROC-ESM-CHEM model has the largest difference between
G4 and RCP4.5 (-16%) while CanESM2 shows the smallest difference (-0.3%). The
time series indicate that tropical storms will become more frequent with time and that
G4 significantly reduces the numbers.
Fig. 1 also shows the evolution of ventilation index in the TC seasons during 2020
to 2069 among the six models. Note that following the definition of VI in Tang et al.
(2014) we use the median value not its mean. During most years from 2020 to 2069,
CanESM2, HadGEM2-ES, MIROC-ESM-CHEM and NorESM1-M show the VI under
G4 lies above that under RCP45. There are no significant trends throughout the period
though all models show slight decreasing trends. Ventilation is disadvantageous for TC
genesis. Thus, reducing trends suggest more storms in future, consistent with trends in
GPI. As with GPI there is about a factor of 2-3 range in absolute values between the
models.



Fig. 2 shows that the correlations between model differences G4-RCP4.5 for annual
GPI and VI. Most models show significant anti-correlation across all TC basins, with
the ensemble having significant anti-correlations for all TC basins except South Pacific.
The degree of correlation varies widely across the models, with some having
coefficients at great as -0.7 and others as low as 0.1. The ensemble mean correlation is
only around -0.25, indicating that GPI and VI are addressing sufficiently different
aspects of TC to warrant independent analysis.
We next examine the spatial pattern of GPI and VI calculated over the 30-year
period: 2040–2069 in the G4 and RCP4.5 experiments. The relative differences as
percentages $(GPI_{G4}-GPI_{RCP4.5})/GPI_{RCP4.5}$ during the peak 3-month season of each
hemisphere's TC season are shown in Fig. 3. These geographic patterns can be
compared with the values in Table 3.
Fig. 3a shows that the GPI anomaly varies by region and by model. For instance,
all models except NorESM1-M show negative differences in the North Indian basin. In
the Western North Pacific, all models except CanESM2 and HadGEM2-ES show
negative differences. Negative differences indicate fewer tropical storms with
geoengineering than under greenhouse gas forcing alone. Despite model differences,
the ensemble result shows robustly that the GPI difference generally negative in the
northern hemisphere but positive in the southern hemisphere. At present the vast
majority of tropical storms occur in the northern hemisphere (Table 2), so the overall
global numbers would likely decrease.



The spatial distribution of VI also has large variation (Fig. 3b). In the West North
Pacific, all models except MIROC-ESM and BNU-ESM have increases, suggesting
fewer cyclones in agreement with the results of GPI. All six models have increases in
the North Atlantic. In the North Indian Ocean, all models show increasing ventilation
index except MIROC-ESM-CHEM and NorESM1-M models, but in the South Indian
Ocean, BNU-ESM model shows a decrease, while other models increase. The ensemble
results are similar as GPI except for the North Indian basin.
**3.2 Accounting for changes in GPI and VI**
We use two different methods to examine how the contributing climate variables to
GPI and VI account for differences between models and across the TC basins. The
objectives are 1) learn which are the key variables in the model simulations of cyclones;
2) find a subset that can be tested against the understanding of how aerosol injection
affects the atmosphere heat and water balance and 3) examine if variations in TC basin
extent or cyclone seasons may be expected under aerosol injection.
**3.2.1 Monthly differences in GPI and VI components between G4 and RCP4.5**
To examine the effects of geoengineering on cyclone seasonality, we look at the
monthly contributions of the factors that make up GPI and VI. We can express Equation
(1) for GPI as the product of four items, respectively representing an atmospheric
absolute vorticity item (*AV*), a vertical wind shear item (*WS*), a relative humidity item
(*RH*), and an atmospheric potential intensity item (*PI*).





$$GPI = \frac{PI \times RH \times AV}{WS} \qquad (5)$$

Where $PI = \left(\frac{Vpot}{70}\right)^3$, $RH = \left(\frac{H}{50}\right)^3$, $WS = (1 + 0.1V_{shear})^2$, $AV = |10^5\eta|^{\frac{3}{2}}$.
The absolute vorticity and vertical wind shear items can be considered to be
dynamic components, while the relative humidity and potential intensity items are
thermodynamic ones.
We follow Zhi et.al. (2013) in identifying the individual monthly contributions
from the four large-scale environmental processes. First taking the natural logarithm of
both sides of Eq. (5), obtains
$\log(GPI) = \log(PI) + \log(RH) - \log(WS) + \log(AV) \qquad (6)$
And differentiating yields
$$\frac{dGPI}{GPI} = \frac{dPI}{PI} + \frac{dRH}{RH} - \frac{dWS}{WS} + \frac{dAV}{AV} \qquad (7)$$

Substituting Eq. (5) into Eq. (7), we have
$dGPI = dPI \times \frac{RH \times AV}{WS} + dRH \times \frac{PI \times AV}{WS}$
$\qquad - dWS \times \frac{PI \times RH \times AV}{WS^2} + dAV \times \frac{PI \times RH}{WS} \qquad (8)$
Eq. (8) can be expressed as annual means and monthly anomalies:
$\delta GPI = \alpha_1 \times \delta PI + \alpha_2 \times \delta RH + \alpha_3 \times \delta WS + \alpha_4 \times \delta AV \qquad (9)$



Where

$$\alpha_1 = \frac{\overline{RH} \times \overline{AV}}{\overline{WS}}$$

$$\alpha_2 = \frac{\overline{PI} \times \overline{AV}}{\overline{WS}}$$

$$\alpha_3 = -\frac{\overline{PI} \times \overline{RH} \times \overline{AV}}{\overline{WS}^2}$$

$$\alpha_4 = \frac{\overline{PI} \times \overline{RH}}{\overline{WS}}$$

And    $\delta GPI = GPI - \overline{GPI}$
In Eq. (9), a bar denotes an annual mean value, and $\delta$ represents the difference between
an individual month and the annual mean, assuming constant coefficients for $\alpha_1$, $\alpha_2$,
$\alpha_3$, and $\alpha_4$.

We are interested in detecting changes between greenhouse gas forcing alone and

under geoengineering, so we examine the differences G4-RCP4.5 for each model
grouping the TC basins by hemisphere in Fig. 4, and use $\delta GPI_{G4} - \delta GPI_{rcp45}$ to
calculate the difference. Fig. 4 clearly shows that *RH* and *WS* make the largest
contribution to GPI differences in both hemispheres in all models except MIROC-ESM-
CHEM. In the Northern Hemisphere, *RH* and *WS* items show negative contributions in
the cyclone season. Hence, these are the factors that enables geoengineering to reduce
GPI relative to greenhouse gas forcing. In the Southern Hemisphere there are no clear
difference between GPI under G4 or RCP4.5. Absolute vorticity, *AV* makes almost no
contribution to the GPI differences under geoengineering in all models.

We also do the same mathematical transform for ventilation index. We obtain

annual means and monthly anomalies:



$$\delta\text{VI} = \alpha_5\delta(V_{pot}) + \alpha_6\delta(\chi_m) + \alpha_7\delta(V_{shear})$$ (10)
Where $\alpha_5 = -\overline{V_{shear}}\dfrac{\overline{\chi_m}}{\overline{V_{pot}^2}}$ $\alpha_6 = \dfrac{\overline{V_{shear}}}{\overline{V_{pot}}}$ $\alpha_7 = \dfrac{\overline{\chi_m}}{\overline{V_{pot}}}$
$$\delta\text{VI} = \text{VI} - \overline{VI}$$
Analogously as for GPI, we show also results for VI in Fig. 4. $V_{shear}$ makes the
largest contribution to ventilation index differences between geoengineering and
greenhouse gas forcing in both hemispheres. Fig. 4 shows that the HadGEM2 values
tend to be smaller than for other models and often differ in sign of difference from the
other models, consistent with the muted spatial patterns in Fig. 3.
**3.2.2 Contributions to GPI and VI across TC basins**
The GPI and VI dependencies may be expressed as a regression equation of $X$ on $Y$
where $Y$ is the GPI or VI anomalies under G4 relative to RCP4.5, and the fractional
contribution to variance, $S$, of each variable $i$ in $X$ to $Y$ can be written, following Moore
et al. (2006) as,
$$S_i = M_i C_i \sigma X_i / \sigma Y$$ (11)
where the $\sigma X$ are the standard deviations of the predictor terms, $\sigma Y$ is the standard
deviation of the anomalies, $C$ are the correlation coefficients of the $X$ with $Y$, $M$ are the
regression coefficients of the $X$ with $Y$. The regression can be expressed as a multiple
linear regression in log space, and the coefficients simply transformed after fitting.
Fitting in log space also allows for the generally heteroscedastic fractional nature of the
errors in the variables.





The relative contributions to GPI anomalies from its four variable items following
the regression Eq. (11) are shown in Fig. 5. *RH* is the dominant factor for GPI
differences in all models except MIROC-ESM-CHEM and all TC basins. A striking
feature of Fig. 5 is that there are very similar patterns of variability between models
across all the basins for the *PI* and the *RH* terms, but not for the *WS* and *AV* terms. Fig.
5 also shows that *AV* makes very little contribution to variance explained in the (G4-
RCP4.5) differences. For all models except MIROC-ESM-CHEM, *WS* makes about
half the contribution to variance explained as *RH*.
Fig. S1 shows the three variables of the ventilation index in a similar way as Fig.
5. $V_{shear}$ makes the largest contribution to VI for all TC basins and all models
especially for the BNU-ESM and MIROC-ESM models. Indeed from Fig. S1 it appears
that VI may be simply replaced by $V_{shear}$, but viewing the month by month
contributions in Fig. 4 shows that other components are relatively important for some
models during some months of the TC season. $\chi_m$ has no consistent contribution for
the models and basins, and it sometimes make negative contributions to the difference
(GPI$_{G4}$-GPI$_{RCP4.5}$).
The statistical power of a regression equation can be expressed as the F-statistic.
Given that the different variables in Figs 5 and S1 show notable differences in their
contribution to the GPI and VI, we can use the F-statistic to examine if a reduced model
with fewer variables is a better statistical model for the differences under G4 and
RCP4.5. GPI has four variables, so there are 15 combination to examine as shown in





Fig. 6. Only for BNU-ESM and MIROC-ESM do the full set of variables have the
highest F-statistic. NorESM1-M and MIROC-ESM-CHEM stand out as different from
the other models in their general behavior. MIROC-ESM-CHEM is largely governed
by *PI* and NorESM1-M by *RH*. In general the models show *RH* has the largest F-statistic
for single parameter models, consistent with Figs. 4 and 5. VI has 3 variables, so there
are 7 combinations possible. Fig. S2 shows $V_{shear}$ has largest contribution to VI for
most of models, and as for GPI, only BNU-ESM and MIROC-ESM models have largest
F-statistic for the full set of model variables.
**3.3 The key factors affecting TCs**

The analysis above shows that *PI* is an important factor affecting TC genesis.

According to Eq. (2), $V_{pot}$ is dependent on the static stability of the troposphere, which
is related to both sea surface ($T_S$) and upper tropospheric temperatures ($T_O$) where
rising air flows out of the storm, and which can be represented by tropical tropopause
(100 hPa) temperature. Fig. S3 show the correlations across TC basins and seasons for
the various fields in RCP4.5 and G4, while Fig. 7 shows the correlations in the
differences between G4 and RCP4.5 so that difference made by the geoengineering can
be clearly evaluated. Fig. 7a shows the dependence of $V_{pot}$ differences (G4-RCP4.5)
on ($T_S$ -$T_O$) differences for the models. All models have significant correlation for all
TC basins except BNU-ESM, which is significant in WNP, ENP, NI and integrated over
all TC basins. However, there is an even stronger dependence for $V_{pot}$ on $T_S$
anomalies (Figs. 7b, S3). The model ensemble $V_{pot}$ is better correlated with $T_s$ rather



than $(T_S - T_O)$ mostly due to better correlations of NorESM1-M and HadGEM2-ES in
Fig. 7b. Fig. S3 shows that correlations for both models under RCP4.5 and G4
separately are not atypical, simply that their (G4-RCP4.5) differences are small. It is
also notable that there are worse correlations for the model ensemble values of $(T_S - T_O)$
with $V_{pot}$ under G4 than RCP4.5 (Fig. S3). All models except CanESM2 and NorESM1
show significant correlation between GPI and $T_S$ anomalies shown as Fig. 7c. And all
except these two models have significant correlations for all TC basins.
Figs. S4 and S5 show the seasonal variability of $T_s$ and $T_0$ for all the models. The
annual cycle of $T_s$, is very similar, as expected for all the models, with good agreement
on the differences in seasonal cycle between the Northern and Southern Hemispheres.
However, for $T_0$ the models show differences in the shapes and phases of the cycles in
both hemispheres, for example only the NorESM1-M model shows roughly antiphase
seasonality between the hemispheres. Fig. S6 shows the ERA-interim reanalysis $T_0$ data,
which has similar seasonality in both hemispheres, with peak temperature anomalies in
August (~ 1.5°C) and a sharp decline to a long minimum by November or December of
similar magnitude. Comparing Figs. S5 and S6 shows that the models generally follow
similar patterns under both G4 and RCP4.5, except for NorESM1-M and HadGEM2-
ES. HadGEM2-ES is also the model with largest amplitude of seasonal cycle, somewhat
larger than in ERA-Interim; other models have smaller amplitudes, with many around
half that observed at present. This degree of difference in $T_0$ simulation likely explains
much of the inter-model differences in GPI.



The other common factors across models and basins that affect TCs are relative
humidity ($H$) and vertical wind shear ($V_{shear}$). In Figs 7d and 7e we plot $H$ and $V_{shear}$
differences between G4 and RCP4.5 as a function of sea surface temperature differences.
Relative humidity rises with warming temperatures under both G4 and RCP4.5 (Fig.
S3), as expected. But there are obvious differences across the ocean basins with weakest
response in ENP, NA and NI and strongest correlations in the Southern Hemisphere
basins. Differences G4-RCP4.5 follow a similar spatial pattern, but with a significant
anti-correlation in North Atlantic. Across-model differences are larger for correlations
of $V_{shear}$ and $Ts$ under both G4 and RCP4.5 (Fig. S3) than for the other key variables. In
contrast with the other parameters, there is generally an anti-correlation with $T_s$ across
all ocean basins, with the NA basin having the weakest correlations. In terms of the
differences in Fig. 7e, all models show clear significant anti-correlations except
CanESM2, with the NI and NA basins having weakest correlations. Vecchi et al. (2007)
found the tropical Atlantic wind shear increases in model projections under global
warming. If the models here capture the effect under G4 and RCP45, we would expect
positive correlation between $V_{shear}$ and $T_s$ over the tropical Atlantic for G4 and RCP4.5
in Fig. S3, but all models show negative correlations, although the Pacific Ocean basins
more significantly anti-correlated than NA. Li et al. (2010) showed that under warming
there is relative shift of towards the central Pacific Ocean of TC genesis away from the
North West Pacific. When we plot the G4-RCP4.5 GPI difference map over the Pacific
Ocean, we also see a clear anomaly in the Central Pacific (Fig. S7). Li et al. (2010)
showed the same effect when using prescribed sea surface temperature patterns from a



suite of models, and they account for the changes in TC by surface temperature
gradients that drive trade winds, which changes the wind shear. Our result is thus
consistent with their findings of changes under greenhouse gas forcing in the Pacific
Ocean if the G4 simulation reverses the effects of RCP4.5 effectively.
**3.4    The effect of ENSO on GPI**

The El Nino-Southern Oscillation (ENSO) is characterized by interannual sea

surface temperature (SST) variations in the eastern and central equatorial Pacific Ocean.
The impact of ENSO events on the TC activity over the western North Pacific (WNP)
has been studied to provide a better understanding of the large-scale steering flow of
TCs and the tendency of TC tracks to shift (Wang et al., 2002). There is also clear
evidence of teleconnections between ENSO and North Atlantic hurricane season
statistics (Gray, 1984; Grinsted et al., 2013). ENSO may be characterized by measures
of atmospheric or oceanic variability. We examined the simulated Niño3.4 index of
tropical Pacific SSTs in the box 170°W - 120°W, 5°S - 5°N, and the Southern
Oscillation Index (SOI) of standardized sea level pressure differences between Tahiti
and Darwin, Australia. Previous analysis of the GeoMIP model ENSO response
(Gabriel et al., 2015) preferred SST based estimates than noisier atmospheric
representations. They also excluded the BNU-ESM, MIROC-ESM and MIROC-ESM-
CHEM models from their analysis because of the model's unrealistic amplitudes of
ENSO. However, as in the real world, all models and the ensemble we use, show a
significant anti-correlation between Niño3.4 index and SOI, except NorESM1-M under



G4, (Fig. 8). This suggests that while many models, are deficient in aspects of their
ENSO variability, they all capture at least some important aspects of ENSO. The
correlation coefficients are more significant in RCP4.5 than under G4 for most models.
We combined Niño3.4 and SOI indices with equal weighting to get a single
representative index of ENSO to compare with GPI and VI.
Annual GPI for the TC basins and the ENSO index during the TC seasons are, in
general, significantly correlated under both G4 and RCP4.5 (Fig. 9). The exception
being CanESM2 which exhibits anti-correlation between GPI and ENSO index under
both G4 and RCP4.5. The analysis for individual basins indicates most models have
significant correlations with ENSO in the WNP and the SP basin, except CanESM2
under the G4 experiment, where it is significantly anti-correlated for RCP4.5. BNU-
ESM, MIROC-ESM and MIROC-ESM-CHEM have significant correlations in ENP,
with NorESM1 and CanESM2 having little or no correlations. Only MIROC-ESM-
CHEM has significant correlation between GPI and ENSO in the NA basin, but the $R^2$
is relatively low, around 0.22. Both BNU-ESM and NorESM1 have significant
correlations in the SI basin, while CanESM2 has significant anti-correlation there. So
the impact of ENSO is most consistently felt in the Pacific Ocean, with perhaps
surprisingly low correlation in the North Atlantic considering the well-known
teleconnections with hurricane activity there.
**3.5 TC from Track with HadGEM2-ES**





As a supplemental analysis to the results based on the GPI and VI, we also employ
a widely-used feature tracking software (TRACK vn. 1.4.9) to directly track vorticity
maxima that characterize cyclones. Hodges (1995) provides a detailed account of
TRACK's core functionality. Jones et al. (2017) also used TRACK to assess
geoengineering impact on North Atlantic hurricane statistics, and we follow their
approach. Firstly, we determine the relative vorticity ($\xi$) on the 850, 500, and 250 hPa
vertical pressure levels from the zonal ($U$) and meridional ($V$) wind using the definition:
$\xi = (1/a \times \cos(\theta)) \times (dV/d\lambda - dU\cos(\theta)/d\theta)$, where $a$ is Earth's radius, and $\theta$ and $\lambda$ are the
latitude and longitude in radians respectively. $U$ and $V$ are required on 6 hour time steps,
but are only available for the HadGEM2-ES model in our ensemble, and limited to the
Northern Hemisphere TC season. TRACK detects storms lasting at least 2 days and
additionally requires values setting for three parameters. We follow Jones et al. (2017)
in selecting: $\xi_I \geq 4.5$ to express the minimum vorticity intensity required; $\xi_V \geq 3.5$ for
the warmth of cyclone core; $\xi_I$ and $\xi_V$ thresholds must be met for at least 4 consecutive
time steps. These criteria represent a relaxation of standard parameters (6, 6, 4) but were
tuned to produce a match in the statistics of Atlantic hurricanes contained in the
HURDAT2 database (Landsea,et al., 2013) from the HADGEM2-ES historical
simulation.
In contrast with Jones et al. (2017) which used data from June through November,
we confine the analysis to the Northern Hemisphere TC season (August, September,
October). The TRACK results suggest that there are significantly more TC under G4
than with RCP4.5 (Table 4) in all basins except the Eastern North Pacific. This



surprising result is not consistent with the changes in GPI and VI for the Northern
Hemisphere (Table 3). Table 3 shows that the G4 cools relative to RCP4.5 and that wind
shear increases. Furthermore, the TRACK result is not consistent with i) the findings of
the statistical model based on surface temperatures (Moore et al., 2015), ii) the proxies
(including wind shear) for TC examined by Jones et al. (2017), iii) the statistical-
dynamical downscaling CHIPS model of Emanuel (2013). Jones et al. (2017) show that
TCs numbers evaluated using the direct counting of storms using the TRACK scheme
(Bengtsson et al., 2007) produce much smaller differences between G4 and RCP4.5
than those using statistical downscaling based on either statistical-dynamical
downscaling using CHIPS (Emanuel et al., 2004) or simply surface temperatures
(Moore et al., 2015).
**4 Discussion and Conclusion**
Typical ESM are run in coarse-resolution that cannot resolve tropical cyclones and
hence do not directly reproduce observed storm intensities and synoptic features related
to cyclogenesis (Camargo, 2013). The storms that may be counted using indirect
methods such as the TRACK algorithm include the whole climate condition. Statistical
methods (Moore et al., 2015) also implicitly include feedbacks between storm and
background climate conditions, but dynamical downscaling methods (Emanuel, 2013)
cannot include them. The GPI and VI proxies we apply here are useful tools for relating
storm activity to meteorological conditions but do not account for changes to TC tracks
or intensity. Since they require relatively little data to calculate (monthly means),





compared with daily or 6 hourly data required for TRACK or the CHIPS tools, and they
convey information from more than simply surface temperature fields, they may give
reasonable insights into the complex changes to TC under SRM geoengineering
schemes.
We evaluated the hurricane index over six TC ocean basins in six CMIP5 and
GeoMIP models. We used G4 and RCP4.5 experiments to assess and compare the
genesis potential and ventilation indices that diagnose tropical storms in climate models.
Based on the climatology of the years 2040-2069, GPI and VI both show small rising
trends for TC genesis in all six models under both G4 and RCP4.5 scenarios. Spatial
patterns of TCs, show both GPI and VI predicting fewer TC in the North Atlantic and
North Indian Ocean under G4 compared with RCP4.5, and more TC in the South Pacific
for most models in the ensemble. Thus stratospheric sulphate aerosol injection could
lead to fewer TCs in the North Atlantic and Indian Ocean but more TCs in the South
Pacific region than under greenhouse gas induced global warming. There is, however,
large inter-model variations across the six ocean basins. The impact of ENSO on TCs
can be detected in the GPI and shows a rising tendency for GPI under El Niño
conditions across the TC basins, especially in the Pacific Ocean.
Detailed statistical analysis of the two TC indices indicates that the thermodynamic
variables potential intensity and relative humidity are the dominant ones affecting
genesis potential, while the dynamic variables such as absolute vorticity and entropy
deficit are much less important. Vertical wind shear is a dynamic variable and dominates
the ventilation index. By examining the contributions of variables to differences in GPI



and VI under geoengineering and greenhouse gas forced climates we show that relative
humidity is the dominant factor for GPI differences in all models and all TC basins,
except MIROC-ESM-CHEM for which potential intensity is the dominant factor. The
analysis suggests that a simplified representation of TCs depending on fewer variables
is possible, but does require analysis of particular model behavior before choosing those
variables. Although wind shear is important and a dynamic variable, it in encouraging
that the thermodynamic state of the system is of prime importance for the GPI,
suggesting that statistical methods of predicting changes in hurricane and storm
behavior are plausible. But, these indices cannot fully represent the actual TC variations
due to the complexity of TC genesis and evolution.

Potential intensity is related to the difference between sea surface temperature and

outflow temperature (the 100 hPa level). In fact we note that SSTs alone provide a better
correlation with both potential intensity and GPI. This result is similar with previous
observational (Grinsted et al., 2013) and modeling (Wu and Lau, 1992) studies that
suggest it is the geographical distribution of SST anomalies that are crucial for the
development of TC. Recent analysis of GeoMIP results by Davis et al. (2016), on the
extent of the tropical belt under G1 and $4 \times CO2$ experiments, demonstrates that tropical
upper-tropospheric temperature changes are well-correlated with the change in global-
mean surface temperature. This is because changes in the static stability characterized
by upper troposphere and surface temperature differences scales with the moist
adiabatic lapse rate and surface temperatures. In contrast with the solar dimming G1



experiments analyzed by Davis et al., (2016), here we analysis G4 which is an aerosol
injection scheme. The aerosol heats the stratosphere mainly between the 16-25 km
elevation injection levels with warming at the tropopause of about 0.6 °C relative to
RCP4.5 for the MIROC-ESM-CHEM model (Pitari et al., 2014). This is about half the
range of the G4-RCP4.5 difference in static stability (Fig. 7). Hence, we would expect
to see a significant improvement in correlation of potential intensity and GPI by using
100 hPa temperatures in addition to SSTs, but we do not. Table 3 shows that the upper
troposphere measured by $T_0$ does not warm with most models under G4, which is
consistent with the impact of G1 on the troposphere. The result is that the models used
here have a better relationship with sea surface temperatures than static stability, and
suggests that the aerosol heating effects are not influencing simulated TC genesis.
The change in relative humidity on the tropical ocean basins in future is a key aspect of
TC genesis according to our analysis. Models tend to agree on the sign of change in
relative humidity as temperatures rise, but there are consistent differences in strength
of response across the ocean basins. The differences in response (G4-RCP4.5) even
indicate a difference in sign of North Atlantic response under geoengineering from the
other basins. This indicates that although relative humidity is important for most models,
changes in TC genesis processes between basins affect its utility as a predictor variable.
The final variable, vertical wind shear, shows large scatter across the models, but
consistent anti-correlation between $V_{shear}$ and surface temperature, and that relationship
is somewhat stronger under G4 than RCP4.5. The changes in GPI over the Pacific



Ocean under G4 compared with RCP4.5 are similar to previous results comparing
patterns of TC genesis under 20$^{th}$ century surface temperatures relative to 21$^{st}$ patterns
(Li et al., 2010). Overall our analysis of the driving parameters in GPI, suggests that
despite large model differences, the simple dependence of GPI on surface temperatures
is reasonably robust.

Smyth et al. (2017) report the seasonal migration of the Intertropical Convergence

Zone (ITCZ) in G1, associated with preferential cooling of the summer hemisphere,
and annual mean ITCZ shifts in some models that are correlated with the warming of
one hemisphere relative to the other. ITCZ location is correlated with tropical cyclone
and season. Our analysis of seasonality of TCs shows that there appears to be a
difference in behavior between the Southern and Northern Hemispheres, with the
southern one showing no consistent changes between models under RCP4.5 and G4
scenarios. Davis et al. (2016) show that there are differences in the evolution of the
northern and southern Hadley cells under greenhouse forcing, with the expansion of the
northern one scaling non-linearly with temperature. Differences seem to be driven
fundamentally by the equator-pole temperature gradient, and therefore may be expected
given the far larger polar amplification in the Northern compared with Southern
Hemisphere.

Many models, owing to their low resolutions, produce much weaker and larger TCs

(Camargo et al., 2005) than seen observationally. Considering the insufficient resolution
of most models, evaluating the GPI and VI may be a better diagnostic of TC variations





under different climates. The results presented here suggest that SRM produces
reductions in TCs across most of the major storm basins, and would be primarily due
to reduced sea surface temperatures in the genesis regions.

***Acknowledgements.*** We thank the climate modeling groups for participating in the
Geoengineering Model Intercomparison Project and their model development teams;
the CLIVAR/WCRP Working Group on Coupled Modeling for endorsing the GeoMIP;
and the scientists managing the earth system grid data nodes who have assisted with
making GeoMIP output available. This research was funded by the National Basic
Research Program of China (Grant 2015CB953600) and the Fundamental Research
Funds for the Central Universities (312231103).



















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



**Tables and Figures**

Table 1. Climate models used in this study

| Model | Reference | Resolution (Lon×Lat) | ensemble members |
|---|---|---|---|
| BNU-ESM | Ji et al. (2014) | 128×64 | 1 |
| CanESM2 | Chylek et al. (2011) | 128×64 | 3 |
| HadGEM2-ES | Collins et al.(2011) | 192×144 | 3 |
| MIROC-ESM | Watanabe et al. (2011) | 128×64 | 1 |
| MIROC-ESM-CHEM | Watanabe et al. (2011) | 128×64 | 9 |
| NorESM1-M | Bentsen et al. (2013) | 144×96 | 1 |


Table 2. Definitions of Regions and numbers of observed TC

| Region | Latitudes | Longitudes | Annual Mean Numbers and percentages (1980-2008) |
|---|---|---|---|
| North Atlantic (NA) | 6-18°N | 20-60°W | 12 (15%) |
| Eastern North Pacific (ENP) | 5-16°N | 90-170°W | 15 (19%) |
| Western North Pacific (WNP) | 5-20°N | 110-150°E | 25 (32%) |
| North Indian (NI) | 5-20°N | 50-110°E | 4 (5%) |
| South Indian (SI) | 5-20°S | 50-100°E | 23 (29%) |
| South Pacific (SP) | 5-20°S | 160E-130°W | |









Table 3. Differences (G4-RCP4.5) in TC basins and season during 2040-2069 year
calculated point-by-point. Northern Hemisphere numbers are above and Southern
Hemisphere below Bold fonts are significant at 95% level. The ensemble means are

not normalized

| Models | $Ts$ (°C) | $To$ (°C) | $Ts$-$To$ (°C) | GPI | $V_{pot}$ (ms$^{-1}$) | $H$ (%) | $V_{shear}$ (ms$^{-1}$) | $\eta$ (×10$^{-8}$ s$^{-1}$) | VI (×10$^3$) | $\chi_m$ (×10$^3$) |
|---|---|---|---|---|---|---|---|---|---|---|
| BNU-ESM | **-0.51** | 0.023 | -0.53 | -0.62 | -0.59 | -0.26 | 0.012 | -1.2 | 20 | 17 |
|  | -0.43 | -0.044 | -0.38 | 0.057 | -0.040 | 0.73 | 0.076 | 0.83 | 7.2 | 19 |
| MIROC-ESM | -0.32 | -0.52 | 0.20 | -0.50 | -1.0 | -0.28 | 0.28 | 1.3 | 15 | -4.9 |
|  | -0.24 | -0.52 | 0.28 | 0.0027 | -0.28 | 0.12 | -0.16 | -0.32 | 1.6 | 8.6 |
| MIROC-ESM-CHEM | **-0.29** | -0.50 | 0.27 | **-2.6** | **6.62** | **4.6** | **1.9** | 0.56 | 8.7 | -11 |
|  | **-0.24** | -0.48 | 0.29 | 0.19 | **6.34** | **3.5** | **2.3** | -0.76 | -5.8 | 1.2 |
| CanESM2 | **-0.50** | -0.13 | -0.37 | -0.017 | -0.86 | 0.17 | -0.045 | -0.08 | 13 | 19 |
|  | **-0.46** | -0.086 | -0.37 | -0.044 | -0.44 | -0.21 | 0.026 | -5.3 | -2.7 | 4.9 |
| NorESM1-M | -0.27 | -0.13 | -0.15 | -2.7 | -1.0 | -0.24 | 0.33 | -3.7 | 19 | 2.8 |
|  | -0.24 | -0.14 | -0.095 | 1.9 | -0.65 | -0.52 | 0.085 | -1.9 | -21 | -9.8 |
| HadGEM2-ES | **-0.75** | 0.13 | -0.88 | -0.30 | -1.2 | 0.43 | 0.083 | 5.8 | 23 | 52 |
|  | **-0.70** | 0.075 | -0.73 | 0.053 | -0.66 | -0.018 | -0.028 | 1.7 | 3.4 | 37 |
| Ensemble | **-0.44** | -0.17 | **-0.24** | -1.1 | **0.33** | 0.73 | 0.43 | 0.44 | 16 | 12 |
|  | **-0.38** | -0.20 | -0.17 | 0.38 | 0.71 | 0.60 | 0.38 | -0.98 | -3.0 | 10 |








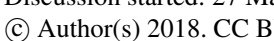




Table 4. Mean TC frequency in Northern Hemisphere basins from the 3-member
ensemble of HadGEM2-ES using TRACK (4.5, 3.5, 4) during August, September,
October 2020-2069. Bold indicates regions with significantly more TC under G4

than RCP4.5 at the 95% level.

| Region | Mean G4 | St.Dev G4 | Mean RCP4.5 | St.Dev RCP4.5 |
|---|---|---|---|---|
| **WNP** | **5.0** | **2.0** | **4.4** | **1.8** |
| ENP | 11.5 | 3.4 | 11.7 | 3.3 |
| **NA** | **1.2** | **1.0** | **0.8** | **0.8** |
| **NI** | **3.5** | **1.5** | **3.0** | **1.7** |












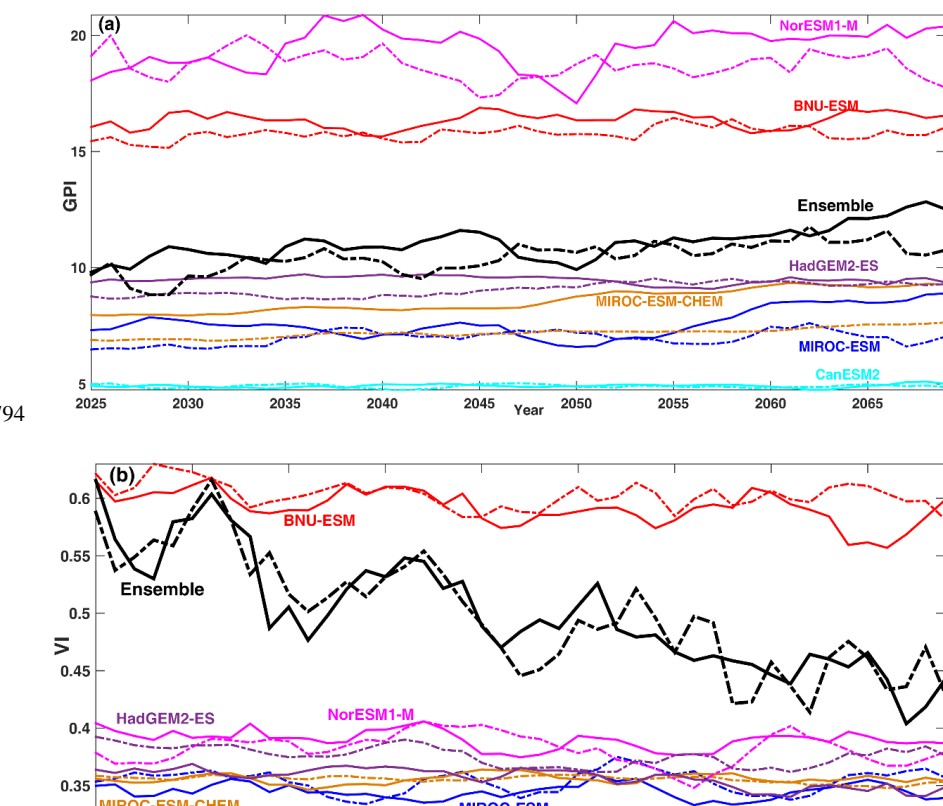


**Figure 1.** Five yearly moving annual averages, of (a) GPI index and (b) ventilation

index in TC season and TC basin. Solid lines denote forcing under RCP4.5 and dotted

lines values under G4. Ensemble mean series were calculate using normalized time

series, shifted by the ensemble mean.



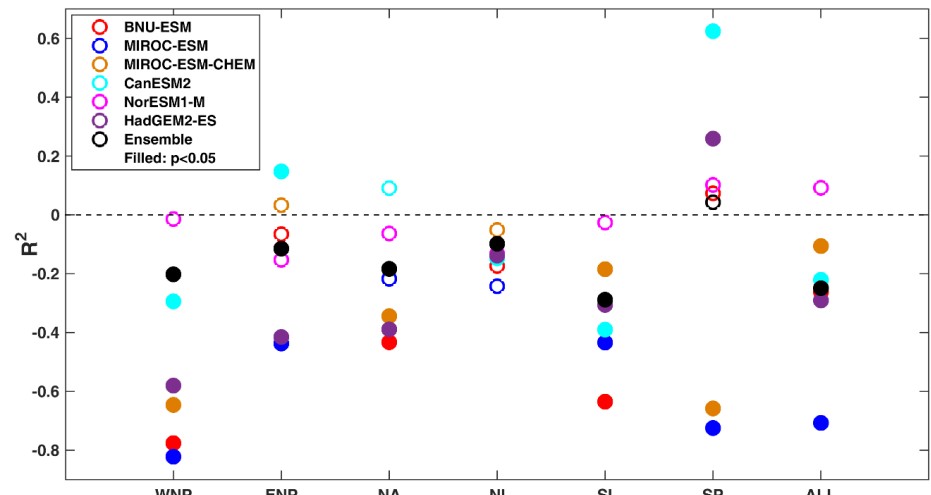

800

**Figure 2.** The correlation coefficients ($R^2$) between annual GPI and VI anomalies (G4-

RCP4.5) during TC season and six ocean TC basins. The MIROC-ESM-CHEM model

has 9 ensemble members, the CanESM2 model has 3 ensemble members, and other

models have one member. Each model is weighted equally and normalized for the

ensemble regardless of the number of separate realizations. Dashed line represent $R^2$=0.



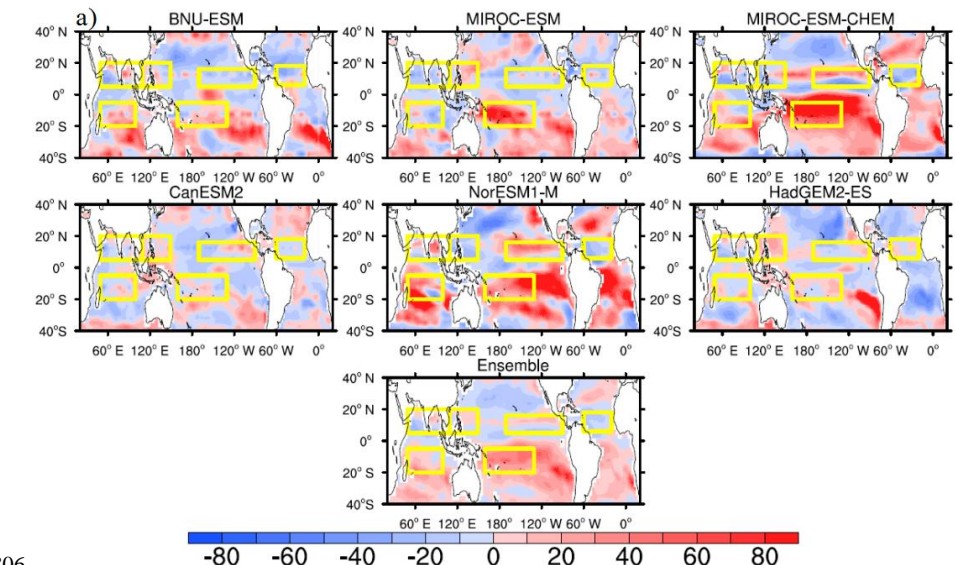


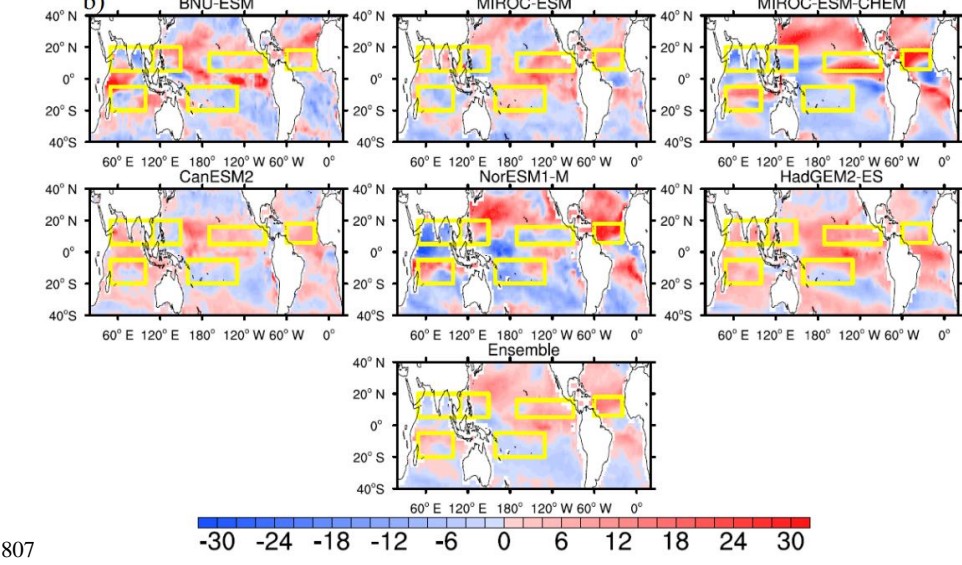


**Figure 3.** Spatial distribution at each grid point during the appropriate TC season

between 2040-2069 of the anomaly $(GPI_{G4}-GPI_{RCP4.5})/GPI_{RCP4.5}$ as a percentage, for a)

GPI and b) VI. Yellow rectangles delimit the six TC ocean basins. The Northern





Hemisphere peak TC season is defined to be August through October, and the Southern
Hemisphere season is defined to be January through March.

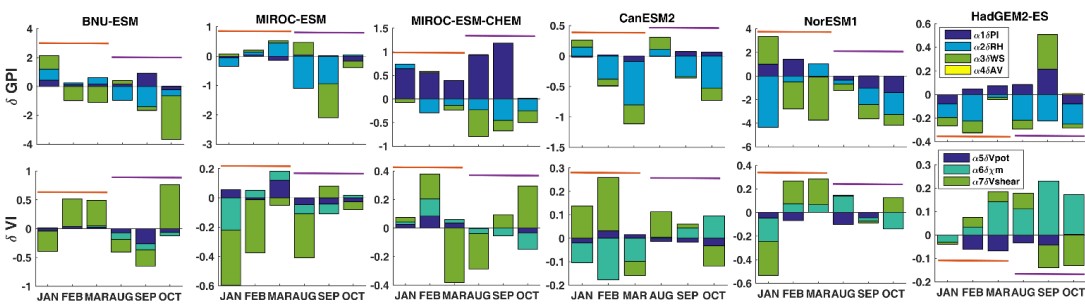

**Figure 4** The mean month contribution of each variable to the difference (G4-RCP4.5)
for the years 2040-2069 in TC basins and TC season in GPI and VI. Brown lines
represent Southern Hemisphere and purple lines represent Northern Hemisphere TC
seasons.






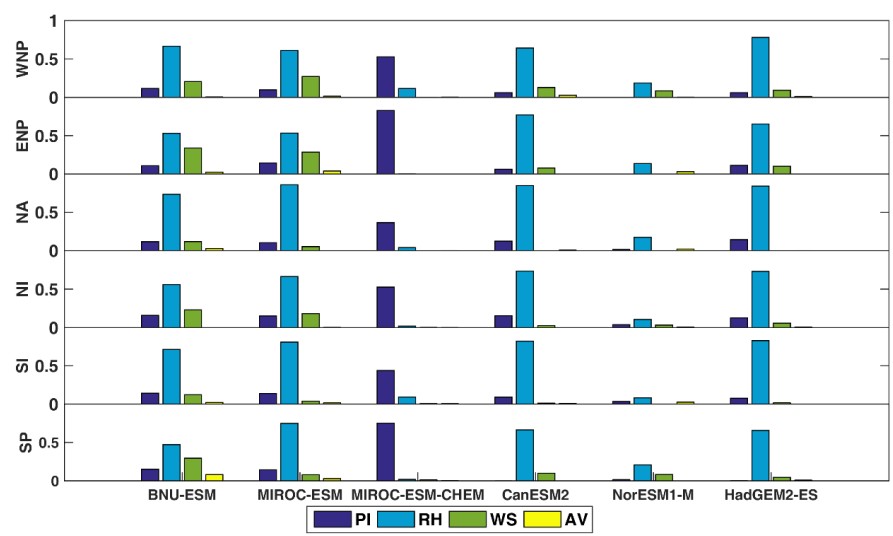


**Figure 5.** The fractional variance contribution of components of GPI during the TC

season and within the six TC basins during 2040-2069.

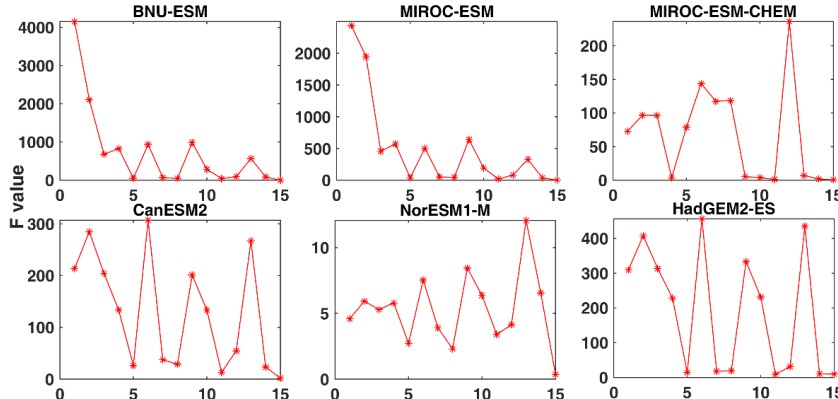


**Figure 6.** The F-statistic of the 15 different combinations of regression variables for

GPI differences between G4 and RCP4.5. The x-axis on each panel represents the

combination of components used as predictors in each regression equation:

1:*(PI,RH,WS,AV)*, 2:*(PI,RH,WS)*, 3:*(PI,RH,AV)*, 4:*(AV,RH,WS)*, 5:*(PI,AV,WS)*,





6:*(PI,RH)*, 7:*(PI,WS)*, 8:*(PI,AV)*, 9:*(RH,WS)*, 10:*(RH,AV)*, 11:*(AV,WS)*, 12:*(PI)*, 13:*(RH)*,
14:*(WS)*, 15:*(AV)*.


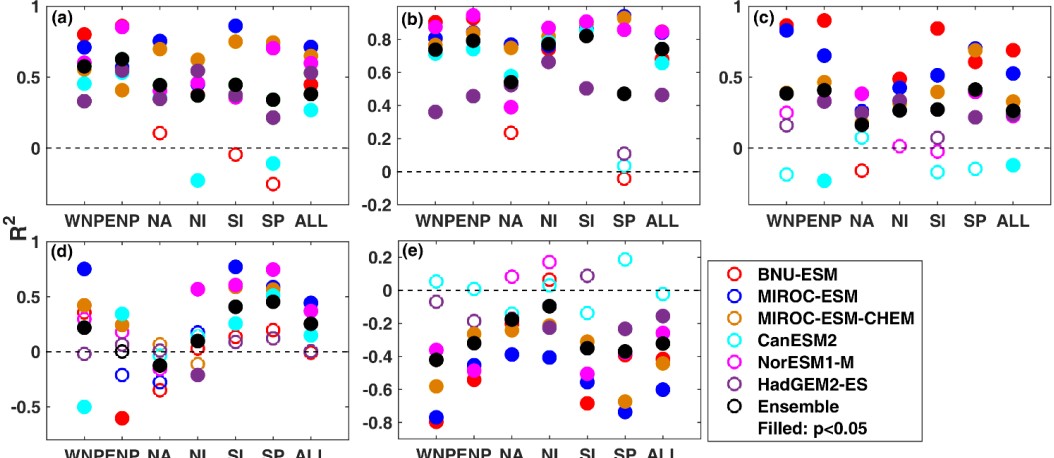

**Figure 7.** The correlations ($R^2$) between differences (G4-RCP4.5) during TC season
and across the six TC basins for the years 2040-2069 for (a) $V_{pot}$ anomalies as a
function static stability $T_S$ - $T_O$. Panels b-e show $R^2$ coefficients for anomalies with sea
surface temperature differences ($T_s$) and: (b)$V_{pot}$, (c) GPI , (d) relative humidity, (e)
vertical wind shear. Each model is weighted equally in the ensembles regardless of
number of observations.





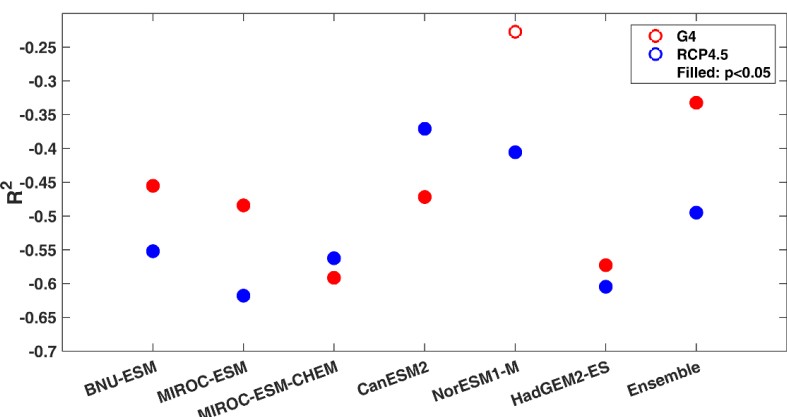


**Figure 8.** The model correlation coefficients between annual Niño3.4 index and SOI

during TC season for 2040-2069.

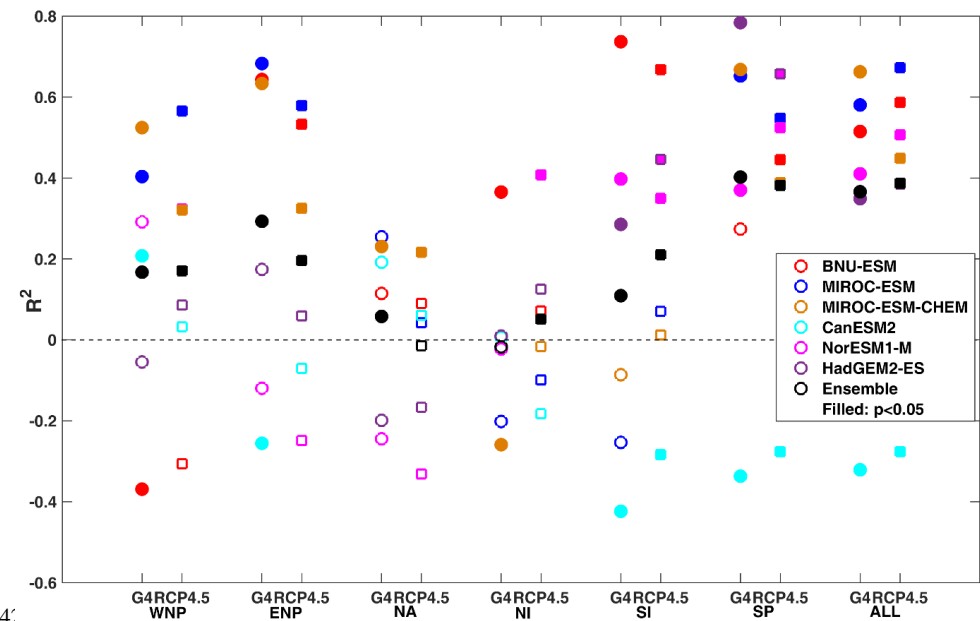


**Figure 9.** The correlation of GPI as a function of (Niño3.4-SOI)/2 during TC season

and six TC basins and all TC basins for the G4 and RCP4.5 experiments.