# Peer review of "Discussion started: 27 March 2018 © Author(s) 2018. CC BY 4.0 License."

_Atmospheric Chemistry and Physics, 2018_

## Referee Comment (RC1) · Anonymous Referee #1 · 12 Apr 2018

**Review of "A statistical examination of the effects of stratospheric sulphate geoengineering on tropical storm genesis" by Wang *et al**

Tropical cyclones (TCs) are high-impact climate phenomena which are poorly represented by the current crop of low resolution climate models. Instead, many studies have utilised implicit statistical relationships between atmospheric conditions and TC activity to investigate changes to TC features under global warming. In this study, Wang *et al* employ the oft-used Genesis Potential Index (GPI) and Ventilation Index (VI) to investigate TC changes under stratospheric aerosol injection (SAI). They find that GPI is lower under SAI than standalone global warming while VI increases, suggesting fewer storms under SAI. They then use a variety of statistical techniques to explore which climatic features contribute to changes in GPI and VI, finding that changes to relative humidity and potential intensity (and furthermore surface temperature) describe most of the variance in the GPI.

I applaud the authors for taking on this hefty subject, and I think they do a commendable job at including many useful analyses. However, I found the manuscript to be poorly written and with a confusing and non-linear structure that reduces readability. In particular, Section 3.5 (TCs from TRACK with HadGEM2-ES) adds nothing to the manuscript and should be removed – especially as it uses only one model (when the manuscript highlights significant inter-model disparities) and a completely different methodology to the rest of the study. I urge the authors to instead focus on the GPI and VI results in Sections 3.1-3.3 which are sufficiently interesting. I'd also suggest adding more justification for the inclusion of Section 3.4, which seems incongruent to the rest of the results section, despite being an interesting study in its own right. Finally, I urge the authors to restructure the Introduction and Conclusion so that key points are not repeated.

I also take issue with fundamental inferences in the manuscript that do not seem to be supported by the tables or figures. Namely, that the GPI (VI) are significantly lower (higher) in G4 than RCP4.5 (L217). In fact, the GPI and VI seem to be very similar between the G4 and RCP4.5 simulations on a model-per-model basis (Fig. 1), and this is also reflected in columns 5 and 10 of Table 3 where only one value is in bold font, suggesting that almost all of the hemispheric differences are not significant. Perhaps add the global-mean GPI and VI differences to Table 3 to confirm that global changes are significant? As this inference (i.e. GPI reduced under geoengineering) is a key result of the paper, I feel this issue must be rectified before I can recommend publication. I have added *many* further comments below.

**Major Comments**

1. I found the abstract to be long and confusing, it should grab the reader with the most significant results. There's lots of irrelevant detail such as "vertical wind shear and vorticity is insignificant" which does not need to be here. The last sentence (Line 52) adds nothing to the abstract and does not follow from the results of the study

2. Restructure the introduction (Lines 83-121). It currently bounces between the various ways of measuring TC activity, when it only needs to say that there are implicit (e.g. GPI), semi-explicit (e.g. dynamical downscaling) and explicit (e.g. feature tracking) methods for measuring storm activity (with references). Then go on to describe GPI and VI theory.

3. I think you need more justification in the text as to why Section 3.4 belongs in the manuscript – it seems to me to be a rather unnecessary accessory that detracts from your GPI and VI results. Also, are there any references that have identified a clear physical relationship between ENSO and storms outside the WNP and NA basins?

4. Remove Section 3.5 which is confusing and does not anything to the study

5. Please establish that the GPI and VI differences between G4 and RCP4.5 are significant as Table 3 and Figure 1 do not currently support this argument

6. There are *many* grammatical and spelling issues that need addressing. Please have the study checked for grammar.

**General Comments**

1. Use acronyms throughout the manuscript – please stop jumping between using acronyms and full terminology (e.g. potential intensity and PI are used interchangeably) which is confusing. Define acronyms on first usage (e.g. Greenhouse Gas GHG) and revert to using them exclusively

2. Use either Stratospheric Sulphate Geoengineering (SSG) or Stratospheric Aerosol Injection (SAI) throughout instead of generic terms such as 'geoengineering' or 'SRM' which comprise a variety of other methods that may have completely different impacts on storms

3. I'd suggest either using Tropical Storms (TS) or Tropical Cyclones (TC) as the terminology throughout. I see you use typhoon or hurricane at some points, which are basin-exclusive terms and I think not be used

4. Occasionally you refer to the use of climate indices such as GPI or VI as the 'direct' way of measuring TC activity (e.g. L85) or you say storm tracking is 'indirect' (L489). I disagree completely! Rather, tracking storms is more direct or explicit, whereas indices are implicit or as you say empirical. Please change this throughout the manuscript.

5. You often give p-values – note within the text which tests were used to derive these p-values (I assume 2-sided t-test but this should be specified)

6. Please check references throughout. My particular gripes are that all papers with 2 authors should be labelled as such, for instance 'Tang and Emanuel (2012)' not 'Tang et al. (2012)' (see L100). I found *many* such instances. Some references are misused and do not contain pertinent detail to the text (see specific comments for details), while the Thomas et al. (2015) reference (Line 120) is missing.

7. If you do keep Section 3.5 in the manuscript, please acknowledge the Hadley Centre for providing you with the 6 hourly data. Please also acknowledge Kevin Hodges if he assisted you in running TRACK.

**Specific Comments**

1. L33 – 'a complete description of TC variability requires much more dynamical data than models can provide at present' – I don't think this is true, I think the issue is the coarse spatiotemporal resolutions, the models have sufficient dynamics. Please rephrase.

2. L35 – do you need to list all the individual components of the GPI and VI in the abstract? Surely this is more for the Introduction or Methods section

3. L41 – 'Globally, GPI under G4 is lower than under RCP4.5, though both have a slight decreasing trend'. I am concerned that people might read from this that SAI is not able to counteract GPI changes under global-warming. Rather, the slight decreasing trend in G4 simply relates to the experimental design (i.e. a constant forcing). I would remove the 'slight decreasing trend' line and add a caveat in the conclusions saying if a different SAI approach were taken (e.g. Jones et al (2018) stabilizing global warming at 1.5K) then the GPI trends may be different

4. L42 – 'spatial patterns in the effectiveness of geoengineering show reductions in TC' – I'm not sure what this means, please clarify

5. L47 – 'genesis potential' -> 'GPI'

6. L52 – final line – again I'm not sure what you mean by this final sentence. Do you mean that simple statistical models based on surface temperature or relative humidity changes are appropriate for examining TC changes? I don't think you really show this though as you don't explicitly link GPI or VI to modelled TCs in this study

7. L58-L62 – numerous grammatical issues

8. L63 – replace 'retard' with suitable word such as 'counteract'

9. L67 – consider replacing 'facilitate' with 'homogenize'

10. L68 – 'and is supported by about 12 model groups' replace with 'and is currently supported by 12 model groups' – be specific on the number

11. L69 – 'Climate system thermodynamics will certainly change under SRM' – this is a strong statement, change compared to what by the way? If you mean compared to business-as-usual then change (of a kind) may welcome! Please reword

12. L82 – 'methods that rely on the statistical links between the thermodynamics of the ocean and atmosphere with cyclone dynamics have been the topic of studies'. This is not entirely true, Jones et al (2017) do explicitly model storms. Add predominantly between have and been

13. L83 – as mentioned, replace typhoons with TCs

14. L83 – as mentioned in the major comments, this paragraph is very confusing, please revise.

15. L96 – 'factors influence genesis' -> 'factors influence TC genesis' or cyclogenesis

16. L96 – 'a quantitative theory is lacking' – please add a suitable reference

17. L99 – What is the definition of potential intensity, which is rather an abstract concept? Define on first use

18. L108 – Dynamical potential intensity is more about ocean feedbacks (i.e. storms stir up cold water, which in turn reduces the potential intensity) than general ocean impacts

19. L109 – 'These indices represent the climatological thermodynamic spatial and seasonal control -> please simplify, superfluous language

20. L111 – 'more or less beyond the abilities of contemporary climate models'. What about the high-resolution models though, which are able to model storm intensities capably (see Murakami *et al* (2016) and Roberts *et al* (2015))

21. L112 – Wang et al (2012) only consider one basin – please replace with a suitable reference comparing different basins

22. L119 – What do you mean by severe TCs? Perhaps give windspeed constraints

23. L120 – Thomas et al (2015) reference missing from bibliography

24. L120 - Kang et al (2012) reference makes no predictions about future changes in TC activity, please change to a relevant reference

25. L126 – The sentence describing Jones et al (2017)'s results is confusing. All you need to say is the SAI in the north reduces North Atlantic TC frequency, while SAI in the south enhances NA TC frequency. Their results were inconclusive for the G4 scenario, as investigated here

26. L131 – Please sell the merits of your study. No other study has looked at GPI and VI in the context of SAI. No other study has investigated storm changes under SAI in basins outside the North Atlantic! No other study has attempted to attribute changes to storms under SAI to thermodynamic changes. This is good work, an important scientific development, and should be highlighted.

27. L138 – 'We quantify the contribution of each variable to TC genesis using two statistical methods'. This is a rather weak statement, which variables do you study and which statistical methods do you utilize?

28. L139 – 'Finally we study the effect of ENSO on TC and TC track of HadGEM2-ES', what justification have you for including these studies here, they don't fit with the GPI and VI work that you have set up to assess

29. L150-L156 – You present the results from Yu et al (2015) and Moore et al (2015), but these will necessarily differ to your results as you use 6 models (you include NorESM-1) and they use 7 models (CSIRO-MK3L and GISS-E2-R). Please state the temperature changes in *your* ensemble of models, and preferably include ranges. Consider also moving this to the results section of the manuscript

30. L162 – Sentence beginning 'It is, however,' should have a suitable reference

31. L170 – replace 'vector' with 'wind'

32. L172 – define Cp in Equation (2)

33. L174 – why do you use the 100 hPa level for the outflow temperature? Do you have a suitable reference?

34. L177 – please define what the GPI is, i.e. the theoretical maximum intensity, and what increases/decreases to GPI signify (with a suitable reference)

35. L188 - please define what the VI is and what increases/decreases to VI signify (with a suitable reference)

36. L190 – 'greenhouse gas' - > RCP4.5

37. L193 – 'air temperature' on levels or near surface air temperature?

38. L200 – 'researchers' -> 'studies'

39. L200 – Emanuel 2010 is not a suitable reference as it makes no predictions for the future, only studying observations from 1908-1958

40. L200 – 'find' -> 'predict'

41. L201 – 'in the Southern Hemisphere' … under global warming

42. L201 – 'but increasing frequency in the northern hemisphere' – Knutson et al (2015) find no such thing! Sure, they find an increase in the East North Pacific and North Indian basins, but they also predict a decrease in the North Atlantic and West North Pacific basins!

43. L203 – 'The observed TC annual-mean numbers for the period 1980-2008 for each basin are also listed in Table 2' – where did these numbers come from? I can only find the basin boundaries in Emanuel 2010. Are these numbers consistent with your basin boundaries? Please provide a suitable reference

44. L209 – 'annual' -> 'annual-mean'

45. L210 – you use August to October for the Northern Hemisphere, but the North Indian basin has two peaks in activity (one in May) (Li et al., 2013). How does GPI and VI change in this second peak in the Indian basin. Please comment on this.

46. L210 – what percentage of total annual storms in each basin occur during your chosen timeframes? These 3 month timeframes seem very narrow to me

47. L217 – 'Furthermore, the G4 means for all models were significantly lower than their RCP4.5 values' – Table 3 seems to say the opposite, that none of the changes to GPI are significant! Please give annual-mean values, standard deviations and p-values for G4 and RCP4.5, perhaps in Table 1?

48. L221 – 'The time series indicate that tropical storms will become more frequent with time and that G4 significantly reduces numbers' – *Fig. 1 does not indicate this at all to me!* There are many years, for each model, where the GPI is higher for G4 than for RCP4.5. Figure 1 will need rethinking as it does not support the central tenet of your paper. How for instance, can a difference of -0.3 % in CanESM2 be significant?

49. L226 – Sentence starting 'During most years from 2020 to 2069…' – this is hardly a sufficient statistical test for significance, simply saying VI looks higher for G4 than RCP4.5! Please perform significance tests and identify which models show significant VI changes and which ones don't. Table 3 suggests that no VI changes are significant!

50. L231 – 'As with GPI, there is about a factor of 2-3 range in absolute values between the models' – perhaps plot normalised anomalies relative to 2020-2030 in Figure 1 instead?

51. L245 – 'All models except NorESM1-M show negative differences in the North Indian basin' – this may be true on a basin-wide basis, but BNU-ESM shows GPI increases in the Bay of Bengal. This might indicate a change in the spatial distribution of storms in the North Indian basin. This is also apparent in the ensemble-mean

52. L269 – consider changing 'item' to 'component' or 'term' throughout this paragraph

53. L274 – has this decomposition (Eq. 5) been used before? If so, provide a relevant reference

54. L301 – 'Hence, these are the factors that *primarily* enable *solar* geoengineering'

55. L304 – do you have any idea as to why MIROC-ESM-CHEM is so different?

56. L312 – Sentence starting 'Fig. 4 shows that the HadGEM2 values tend to be smaller' – CanESM2 is similarly muted and seems to have the same signs as HadGEM2

57. L356 – 'The key factors affecting TCs' – consider adding a more informative title, possibly, 'Primary factors that control GPI and VI changes'?

58. L357 – You seem to have found in Section 3.2 that relative humidity is the most important factor for GPI, and then ignore this finding in Section 3.3, which I found curious

59. L368 – 'The model ensemble' -> 'The ensemble-mean'

60. L370 – 'Fig S3 shows that correlations for both models under RCP4.5 and G4 separately are not atypical, simply that their G4-RCP4.5 differences are small' – I'm not sure what you mean by this sentence, please rephrase

61. L373 – Similarly the last sentence is unclear. Do you mean that all models excepts CanEMS2 and NorESM1 exhibit significant correlation between Ts and Vpot in all basins?

62. L376 – Change 'variability' to 'cycle' throughout this paragraph

63. L384 – Remove 'Comparing'

64. L388 – The last sentence – can you also plot the seasonal cycle of Ts in ERA-interim just to confirm that all the models are doing reasonably well here?

65. L390 – Consider splitting this paragraph into two. It is too long and unwieldy as it is

66. L391 – Sentence beginning 'In Figs 7d and 7e we plot' – please reword this sentence to something like 'we plot correlations between H / Vshear and Ts.

67. L399 – 'there is generally an anti-correlation *between Vshear and* Ts'

68. L402 – Vecchi *and Soden* (2007) found that wind shear increases in both the North Atlantic *and the East Pacific* under global warming.

69. L404 – 'If the models *assessed* here'

70. L407 to L415 – I'm not sure what you are trying to prove here, it seems peripheral and needs to be reworded

71. L441 – 'The analysis for individual basins indicates most models have significant correlations with ENSO in the WNP' – This is not true! Only 4/7 of the models have significant correlation in the WNP in RCP4.5

72. L448 – is there any previous studies that suggest a link between ENSO and tropical cyclone activity in basins outside the North Atlantic and the Pacific. If so, please cite

73. L449 – 'is most consistently felt in the Pacific Ocean' –particularly the South Pacific

74. L473 – why are the TRACK results so much lower in your Table 4 than in Jones et al. (2017)? For instance, you get 1.2 storms per year in the North Atlantic basin in G4 compared to ~11 per year in their work (their Fig. 4). Their reasoning behind the use of the (4.5,3.5,4) configuration was to attain ~10 storms per year on average in the historical period. Please check these numbers, they seem wrong.

75. L487 – Change 'typical' to 'current'

76. L489 – 'The storms that may be counted using indirect methods such as the TRACK algorithm include the whole climate condition' – This doesn't make sense to me. Consider replacing with 'Simulated storms that may be counted using methods such as the TRACK algorithm allow for feedbacks with the climate system'

77. L490 – 'Statistical methods (Moore et al., 2015) also implicitly include feedbacks between storm and climate conditions' – in what way do they include feedbacks? I don't understand this. They are simply diagnostics

78. L492 – 'but dynamical downscaling methods (Emanuel, 2013) cannot include them' – I disagree, Emanuel employs a simple ocean model which can be adjusted to provide

climate feedback. In fact, I think the semi-explicit scheme offers more opportunity to incorporate feedbacks than the statistical methods

79. L493 – change 'apply' to 'utilize'
80. L495 – change 'relatively little data' to 'coarse temporal-resolution data'
81. L502 – Change 'diagnose tropical storms in climate models' to 'relate tropical storm activity to ambient meteorology'
82. L507 – 'Thus stratospheric sulphate aerosol injection could lead to fewer TCs in the North Atlantic …' – note that this is one solar geoengineering scenario (a uniform one). Injecting aerosol preferentially into one hemisphere may increase the amount of storms in the North Atlantic (Jones et al (2017)) with unknown effects in other basins
83. L510 – 'The impact of ENSO on TCs can be detected in the GPI' – this is poorly worded, you have not explicitly looked at ENSO and TCs, only at ENSO and GPI. Rephrase in such a way: 'ENSO is found to be correlated with GPI'. Are there any implications specifically in terms of solar geoengineering from your results? I mean, is there a decrease in El Nino years in the G4 simulations?
84. L515 – remove 'such as'
85. L521 – 'a simplified representation of TCs depending on fewer variables is possible' -> 'a simplified representation of the GPI depending on fewer variables may be possible'
86. L523 – sentence running from 'it is encouraging that the thermodynamic state …' I don't understand what you mean here?
87. L529 – '(the 100hPa level)' -> (evaluated at 100 hPa)'
88. L529 – Replace 'note that' with 'find that changes to' and add 'changes' after GPI
89. L542 – rather than using temperature changes from Pitari et al (2014), can you give the ensemble mean upper-tropospheric temperature changes from your 6-member ensemble please
90. L542 – 'This is about half the range of the G4-RCP4.5 difference in static stability (Fig. 7)' – Figure 7 does not show that changes to static stability…
91. L544 – remove 'significant'
92. L545 – why does T0 not warm with most models under G4? Do you have a reason that you can offer? It is that the aerosol particles are small?
93. L578 – 'Many models, owing to their low resolutions, produce much weaker and larger TC' – this statement has been repeated a few times (e.g. L487). Please do not repeat statements
94. L582 – change 'would be' to 'this is'

**References**

Jones, AC, *et al.* (2017), Impacts of hemispheric solar geoengineering on tropical cyclone frequency, *Nature Communications*, 8, 1382

Jones, AC, *et al.* (2018), Regional Climate Impacts of Stabilizing Global Warming at 1.5 K Using Solar Geoengineering. *Earth's Future*

Li, M (2013), Bimodal Character of Cyclone Climatology in the Bay of Bengal Modulated by Monsoon Seasonal Cycle, *Journal of Climate*

Murakami, H, *et al.* (2016), Seasonal Forecasts of Major Hurricanes and Landfalling Tropical Cyclones using a High-Resolution GFDL Coupled Climate Model, *Journal of Climate*

Roberts, M, *et al.*, (2015), Tropical Cyclones in the UPSCALE Ensemble of High-Resolution Global Climate Models, *Journal of Climate*

---

## Referee Comment (RC2) · Anonymous Referee #2 · 20 Apr 2018

Review of the paper: *A statistical examination of the effects of stratospheric sulphate geoengineering on tropical storm genesis*, by Q. Wang et al., Atmos. Chem. Phys. Discuss., acp-2018-142, 2018.

In this numerical work, a statistical approach is described for analysing the effects of sulphate geoengineering on the genesis of tropical storms. The procedure is well designed on the general methodology of the GeoMIP project, with use of data that independent global models have provided in a common database with their G4 simulations. The manuscript is scientifically robust and deserves publication on ACP.

Some of the conclusions are important, mainly the fact that the thermodynamic role of SST changes induced by geoengineering aerosols dominates over the lower stratospheric aerosol heating. However, sometimes the authors compare the SST effects with changes in static stability, as if they were two independent things (see for example in the conclusions, lines 547-549). Actually, SST changes may affect the atmospheric static stability by themselves, even in the absence of a stratospheric warming. I would suggest rephrasing. The authors themselves clearly explain how static stability changes are controlled by both surface and upper tropospheric temperatures (page 18, lines 358-360).

This is the main *specific point* I suggest to better clarify all along the manuscript, before final publication on ACP. In addition, it is true that the aerosol heating is mostly located in the 16-25 km layer (see page 27, lines 540-542); however, due to the large size of the geoengineering aerosol particles (effective radius of the order of 0.6 μm or more), a significant fraction of the stratospheric particles would settle down below the tropical tropopause (Niemeier et al., 2010; English et al., 2012; Cirisan et al, 2013), thus producing some diabatic heating superimposed to the convectively-driven upper tropospheric cooling. This means that the surface cooling (with associated upper tropospheric tropical cooling, due to lesser efficient convective motions) may be expected as the dominant process controlling the geoengineering induced changes of atmospheric static stability. At the same time, the aerosol heating in a few kilometres layer immediately below the tropical tropopause (due to gravitational sedimentation of large geoengineering sulfate aerosols) should also be considered as a contributing smaller effect.

It would be worth to note that another indirect effect of sulfate geoengineering, related to the surface cooling and static stability changes, is discussed in Visioni et al. (2018). Here the sensitivity of upper tropospheric ice formation is studied with inclusion of the aerosol-induced surface cooling, with respect to a reference condition documented in Kuebbeler et al. (2016), where only the stratospheric warming due to the aerosols was taken into account. The conclusions presented in the manuscript of Wang et al. (2018) go in the same direction of what discussed in this other study.

*Minor points*

P. 3, line 66: the Kravitz reference has a wrong comma between the name and *et al*.
P. 3, line 72: some more recent articles can be cited here, for example Visioni et al. (2017).
P. 4, line 83: *are* used instead of *have* used.
P. 6, line 126-130: I would suggest rephrasing this concept, maybe splitting the long sentence in two. In its present form it is hard to follow.
P. 7, line 149: explain better the altitude at which the injection is simulated, since it has been shown how different injection heights may affect differently the climate response (Tilmes et al., 2017; Kleinschmitt et al., 2018).

P. 8, line 162-165: for a recent study analysing the connection between the stratospheric warming due to the sulfur injection and the tropospheric response in term of vertical motions, see Visioni et al. (2018) (now under review in ACPD).

P. 14-15: I suggest to the authors to move some of the longer equations derivations to the supplementary material for better readability of the manuscript.

P. 22, line 433: no comma between models and are.

P. 25, line 510: I would suggest using "variability" instead of "variations".

P. 27, line 539: analyze instead of analysis. Better rephrase "aerosol injection scheme" into a more appropriate description, such as "protocol".

P. 27, line 552: "in the response strength across the ocean basins" sounds probably better than "in strength of response across the ocean basins".

P.38, Fig. 1: adding a legend outside the figure, instead of having the names of the models close to the related lines, would make it easier to read.

*References*

Cirisan, A., Spichtinger, P., Luo, B. P., Weisenstein, D. K., Wernli, H., Lohmann, U., and Peter, T.: Microphysical and radiative changes in cirrus clouds by geoengineering the stratosphere, *J. Geophys. Res.-Atmos., 118,* 4533–4548, doi:10.1002/jgrd.50388, 2013.

Kleinschmitt, C., Boucher, O., and Platt, U.: Sensitivity of the radiative forcing by stratospheric sulfur geoengineering to the amount and strategy of the $SO_2$ injection studied with the LMDZ-S3A model, *Atmos. Chem. Phys., 18,* 2769-2786, https://doi.org/10.5194/acp-18-2769-2018, 2018.

Kuebbeler, M., Lohmann, U., and Feichter, J.: Effects of stratospheric sulfate aerosol geo-engineering on cirrus clouds, *Geophys. Res. Lett., 39*, doi:10.1029/2012GL053797, http://dx.doi.org/10.1029/2012GL053797, l23803, 2012.

English, J. M., Toon, O. B., and Mills, M. J.: Microphysical simulations of sulfur burdens from stratospheric sulfur geoengineering, Atmos. Chem. Phys., 12, 4775-4793, https://doi.org/10.5194/acp-12-4775-2012, 2012.

Niemeier, U., Schmidt, H. and Timmreck, C. (2011), The dependency of geoengineered sulfate aerosol on the emission strategy. *Atmos. Sci. Lett., 12*: 189-194. doi:10.1002/asl.304

Tilmes S., Richter J. H., Mills M. J., Kravitz B., MacMartin D. G., Vitt F. Lamarque J.- F. (2017). Sensitivity of aerosol distribution and climate response to stratospheric $SO_2$ injection locations. *J. Geophys. Res.-Atmos.*, *122,* 12,591–12,615. https://doi.org/10.1002/2017JD026888.

Visioni, D., Pitari, G., and Aquila, V.: Sulfate geoengineering: a review of the factors controlling the needed injection of sulfur dioxide, *Atmos. Chem. Phys., 17*, 3879-3889, https://doi.org/10.5194/acp-17-3879-2017, 2017.

Visioni, D., Pitari, G., and di Genova, G.: Upper tropospheric ice sensitivity to sulfate geoengineering, *Atmos. Chem. Phys. Discuss.,* https://doi.org/10.5194/acp-2018-107, under review, 2018.

---

## Referee Comment (RC3) · Anonymous Referee #1 · 21 May 2018

As the title says, I see you have made the corrections but I'd like to read the updated manuscript please - Reviewer 1

———————————————————

---

## Referee Comment (RC4) · Anonymous Referee #2 · 21 May 2018

I would advise the authors to modify some of the new phrases they are adding to the manuscript after my comments in a more correct way. In particular I'm referring to:

"Furthermore, recent work (Visioni et al., 2018 ACP in discussion) explores the secondary of surface cooling on the upper troposphere with the impact on cirrus clouds, and the concomitant impact on static stability. Surface cooling and lower stratospheric warming, together, tend to stabilize the atmosphere, thus decreasing turbulence and water vapor updraft velocities. The net effect is an induced cirrus thinning, which serves to increase net global cooling due to the SAI."

- "explores the secondary of surface cooling on the upper troposphere with the impact

on cirrus clouds" would be much clearer if phrased as such: "explores the surface cooling impact on upper tropospheric cirrus cloud formation".

- "thus decreasing turbulence and water vapor updraft velocities" would be better as "thus decreasing turbulence and updraft velocities".

- "which serves to increase net global cooling due to the SAI" should be changed to "which indirectly increases the net global cooling due to the SAI"
* * *

---

## Author Comment (AC1) · 21 May 2018

In the reply, the referee's comments are in *italics*, our response is in normal text, and quotes from the manuscript are in blue.

***Anonymous Referee #1***

*Major Comments*

*1. I found the abstract to be long and confusing, it should grab the reader with the most significant results. There's lots of irrelevant detail such as "vertical wind shear and vorticity is insignificant" which does not need to be here. The last sentence (Line 52) adds nothing to the abstract and does not follow from the results of the study*

**Reply:** revised as requested

*2. Restructure the introduction (Lines 83-121). It currently bounces between the various ways of measuring TC activity, when it only needs to say that there are implicit (e.g. GPI), semi-explicit (e.g. dynamical downscaling) and explicit (e.g. feature tracking) methods for measuring storm activity (with references). Then go on to describe GPI and VI theory.*

**Reply:** revised as requested

*3. I think you need more justification in the text as to why Section 3.4 belongs in the manuscript – it seems to me to be a rather unnecessary accessory that detracts from your GPI and VI results. Also, are there any references that have identified a clear physical relationship between ENSO and storms outside the WNP and NA basins?*

**Reply:** we deleted section 3.4 as suggested

*4. Remove Section 3.5 which is confusing and does not anything to the study*

**Reply:** we deleted section 3.5 as suggested

*5. Please establish that the GPI and VI differences between G4 and RCP4.5 are significant as Table 3 and Figure 1 do not currently support this argument*

**Reply:** We have done this, making several new tables and figures that show significant changes as tested using Wilcoxon signed rank test and Student's t-test. Fig. 1 has also been revised and the whole results re-evaluated using longer TC seasons.

*6. There are many grammatical and spelling issues that need addressing. Please have the study checked for grammar.*

**Reply:** Done

*General Comments*

1. *Use acronyms throughout the manuscript – please stop jumping between using acronyms and full terminology (e.g. potential intensity and PI are used interchangeably) which is confusing. Define acronyms on first usage (e.g. Greenhouse Gas GHG) and revert to using them exclusively*

**Reply**:In general we have but in some cases such as with the example the referee gives, Potential Intensity, this can lead to confusion. We use *PI* to define a term in equation 5, while potential intensity is defined as $V_{pot}$ in equation 2. This is deliberately done since the equations define different quantities that may be considered as potential intensity in different formulations.

*Use either Stratospheric Sulphate Geoengineering (SSG) or Stratospheric Aerosol Injection (SAI) throughout instead of generic terms such as 'geoengineering' or 'SRM' which comprise a variety of other methods that may have completely different impacts on storms.*

**Reply**:Done.

2. *I'd suggest either using Tropical Storms (TS) or Tropical Cyclones (TC) as the terminology throughout. I see you use typhoon or hurricane at some points, which are basin-exclusive terms and I think not be used*

**Reply**:Done.

*Occasionally you refer to the use of climate indices such as GPI or VI as the 'direct' way of measuring TC activity (e.g. L85) or you say storm tracking is 'indirect' (L489). I disagree completely! Rather, tracking storms is more direct or explicit, whereas indices are implicit or as you say empirical. Please change this throughout the manuscript.*

**Reply**:Done.

3. *You often give p-values – note within the text which tests were used to derive these p-values (I assume 2-sided t-test but this should be specified)*

**Reply**:Done, we use Student's t-tests but usually the Wilcoxon signed rank test.

5. *Please check references throughout. My particular gripes are that all papers with 2 authors should be labelled as such, for instance 'Tang and Emanuel (2012)' not 'Tang et al. (2012)' (see L100). I found many such instances. Some references are misused and do not contain pertinent detail to the text (see specific comments for details), while the Thomas et al. (2015) reference (Line 120) is missing.*

**Reply**:Done.

6. *If you do keep Section 3.5 in the manuscript, please acknowledge the Hadley Centre*

*for providing you with the 6 hourly data. Please also acknowledge Kevin Hodges if he assisted you in running TRACK.*

**Reply:** Section 3.5 is deleted.

*Specific Comments*

1. *L33 – 'a complete description of TC variability requires much more dynamical data than models can provide at present' – I don't think this is true, I think the issue is the coarse spatiotemporal resolutions, the models have sufficient dynamics. Please rephrase.*

**Reply:** rewritten as: an accurate description of TC variability requires much higher spatial and temporal resolution than the models used in the GeoMIP experiments provide

2. *L35 – do you need to list all the individual components of the GPI and VI in the abstract? Surely this is more for the Introduction or Methods section*

**Reply:** rewritten as Genesis potential index (GPI) and ventilation index (VI) are combinations of dynamic and thermodynamic variables that provide proxies for TC activity under different climate states.

3. *L41 – 'Globally, GPI under G4 is lower than under RCP4.5, though both have a slight decreasing trend'. I am concerned that people might read from this that SAI is not able to counteract GPI changes under global-warming. Rather, the slight decreasing trend in G4 simply relates to the experimental design (i.e. a constant forcing). I would remove the 'slight decreasing trend' line and add a caveat in the conclusions saying if a different SAI approach were taken (e.g. Jones et al (2018) stabilizing global warming at 1.5K) then the GPI trends may be different*

**Reply:** Yes, rewritten as GPI is consistently and significantly lower under G4 than RCP4.5 in 5 out of 6 ocean basins, but it increases under G4 in the South Pacific.

4. *L42 – 'spatial patterns in the effectiveness of geoengineering show reductions in TC' – I'm not sure what this means, please clarify*

**Reply:** Yes, rewritten as reply to #3

5. *L47 – 'genesis potential' -> 'GPI'*

**Reply:** Changed.

5. *L52 – final line – again I'm not sure what you mean by this final sentence. Do you mean that simple statistical models based on surface temperature or relative humidity changes are appropriate for examining TC changes? I don't think you really show this though as you don't explicitly link GPI or VI to modelled TCs in this study*

**Reply:** Agreed, we delete the sentence.

*6.   L58-L62 – numerous grammatical issues*

**Reply:** rewritten as: Anthropogenic greenhouse gas (GHG) emissions are changing climate (IPCC, 2007). The best solution for limiting climate change is to reverse the growth in net GHG emissions. It is doubtful that reductions in emissions can be done fast enough to limit global mean temperatures rises to targets such as the 1.5° or 2°C pledged at the Paris climate meeting (Rogelj et al., 2015).

*8. L63 – replace 'retard' with suitable word such as 'counteract'*

**Reply:** Done, replaced with counteract.

*9. L67 – consider replacing 'facilitate' with 'homogenize'*

**Reply:** Done.

*10. L68 – 'and is supported by about 12 model groups' replace with 'and is currently supported by 12 model groups' – be specific on the number*

**Reply:** Done. supported by 15 model groups

*11. L69 – 'Climate system thermodynamics will certainly change under SRM' – this is a strong statement, change compared to what by the way? If you mean compared to business-as-usual then change (of a kind) may welcome! Please reword*

**Reply:** We are not making a value-judgement merely reporting what is well-established, we added a few more references to illustrate the point. Rewritten Climate system thermodynamics will change under SRM geoengineering because the reduction in short wave radiation is designed to offset increases in long wave absorption (Huneeus et al., 2014; Kashimura, H., M. Abe, S. Watanabe, T. Sekiya, D. Ji, J. C. Moore, J.N.S. Cole and B. Kravitz 2017 Shortwave radiative forcing, rapid adjustment, and feedback to the surface by sulfate geoengineering: analysis of the Geoengineering Model Intercomparison Project G4 scenario, *Atmospheric Chemistry and Physics* 17, 3339-3356, doi:10.5194/acp-17-3339-2017, 2017; Visioni, D., Pitari, G., and Aquila, V.: Sulfate geoengineering: a review of the factors controlling the needed injection of sulfur dioxide, Atmos. Chem. Phys., 17, 3879-3889, https://doi.org/10.5194/acp-17-3879-2017, 2017; Russotto, R. D. and Ackerman, T. P.: Energy transport, polar amplification, and ITCZ shifts in the GeoMIP G1 ensemble, Atmospheric Chemistry and Physics, 18, 2287–2305, doi:10.5194/acp-18-2287-2018, 2018. ).

*12. L82 – 'methods that rely on the statistical links between the thermodynamics of the ocean and atmosphere with cyclone dynamics have been the topic of studies'. This is not entirely true, Jones et al (2017) do explicitly model storms. Add predominantly between have and been*

**Reply:** Done.

*13. L83 – as mentioned, replace typhoons with TCs*

**Reply**:Done.

*14. L83 – as mentioned in the major comments, this paragraph is very confusing, please revise.*

Many methods have used to study the changes in TCs under climate warming. These can be divided into implicit methods, such as the GPI and VI which we focus on here, semi-explicit, such as downscaling (Emanuel, 2006; 2013), and explicit such as feature tracking storm systems (Hodges, 1995; Jones et al., 2017). Implicit methods rely on using historical climate and storm records to quantitative relationships between TC and key variables such as local, tropical and global sea surface temperatures, and various teleconnection patterns (Grinsted et al., 2012; Emanuel, 2008; Landsea, 2005; Gray, 1979). Potential intensity theory (Bister et al., 1998; Emanuel et al., 2004) predicts the dependence of TC wind speed on the air-sea thermodynamic imbalance and the temperature of the lower stratosphere. For example, many studies suggest that wind shear has inhibitory effect on the TC activity (Vecchi et al., 2007). Others have also identified changes in the large-scale environmental factors influencing tropical storm activity to assess TC changes in future (Tippett et al., 2011; Grinsted et al., 2013).

*15. L96 – 'factors influence genesis' -> 'factors influence TC genesis' or cyclogenesis*

**Reply**:Changed to factors influence TC cyclogenesis.

*16. L96 – 'a quantitative theory is lacking' – please add a suitable reference*

**Reply**:rewritten as a quantitative theory is lacking (Emanuel, 2013)

*17. L99 – What is the definition of potential intensity, which is rather an abstract concept? Define on first use*

**Reply**:rewritten as The GPI uses four environmental variables: potential intensity, low-level absolute vorticity, vertical wind shear, and relative humidity. Potential intensity is the maximum sustainable intensity of tropical cyclones based on the thermodynamic state of the atmosphere and sea surface, that is the difference between the saturation enthalpy of the sea surface and the moist static energy of the subcloud layer (Riehl H (1950) A model for hurricane formation. J Appl Phys 21:917–925).

*18. L108 – Dynamical potential intensity is more about ocean feedbacks (i.e. storms stir up cold water, which in turn reduces the potential intensity) than general ocean impacts*

**Reply**:rewritten as: Dynamic potential intensity is yet another index designed to describe ocean feedbacks on tropical cyclones, that is storms bring cold deeper water to the surface, which reduces the potential intensity.

*19. L109 – 'These indices represent the climatological thermodynamic spatial and*

*seasonal control -> please simplify, superfluous language*

**Reply**:rewritten as: These indices represent the thermodynamic and hence seasonal control of TC genesis.

*20. L111 – 'more or less beyond the abilities of contemporary climate models'. What about the high-resolution models though, which are able to model storm intensities capably (see Murakami et al (2016) and Roberts et al (2015))*

**Reply**:Yes, a few models can do this, but that's what we meant by "more or less". Rewritten as: which is beyond the abilities of most contemporary climate models, in particular those we use here.

*21. L112 – Wang et al (2012) only consider one basin – please replace with a suitable reference comparing different basins*

**Reply**:We use Emanuel (2010) and Wing et al., (2015) to replace Wang et.al (2012).

*22. L119 – What do you mean by severe TCs? Perhaps give windspeed constraints*

**Reply**:Done, we Rewritten as: the frequency of intense TC (those having windspeeds larger than 55 ms-1)

*23. L120 – Thomas et al (2015) reference missing from bibliography*

**Reply**:Sorry, it should be Knutson et al (2015), and we revised it now.

*24. L120 - Kang et al (2012) reference makes no predictions about future changes in TC activity, please change to a relevant reference*

**Reply:** we delete Kang as 2 other references are already cited here.

*25. L126 – The sentence describing Jones et al (2017)'s results is confusing. All you need to say is the SAI in the north reduces North Atlantic TC frequency, while SAI in the south enhances NA TC frequency. Their results were inconclusive for the G4 scenario, as investigated here*

**Reply:** rewritten as: Jones et al. (2017) showed SAI in the northern hemisphere reduced the numbers of TC in the North Atlantic while SAI in the southern hemisphere increased numbers in the basin.

*26. L131 – Please sell the merits of your study. No other study has looked at GPI and VI in the context of SAI. No other study has investigated storm changes under SAI in basins outside the North Atlantic! No other study has attempted to attribute changes to storms under SAI to thermodynamic changes. This is good work, an important scientific development, and should be highlighted.*

**Reply**:Thanks for this, we rewrite the paragraph as: In contrast with earlier work that has focused only on the impacts of SAI on North Atlantic hurricanes (Moore et al., 2015;

Jones et al., 2017), we examine ESM simulations of global TC evolution in 6 ocean basins using the GPI and VI indices. We then evaluate how far TC changes under SAI and GHG forcing can be attributed to thermodynamic changes, and hence be forecast in statistical terms.

*27. L138 – 'We quantify the contribution of each variable to TC genesis using two statistical methods'. This is a rather weak statement, which variables do you study and which statistical methods do you utilize?*

**Reply**:We rewrite as: We quantify the contribution of SST, relative humidity and wind shear to TC genesis based on attribution of monthly variance in GPI and VI in each basin's time series using multiple linear regression methods.

*28. L139 – 'Finally we study the effect of ENSO on TC and TC track of HadGEM2-ES', what justification have you for including these studies here, they don't fit with the GPI and VI work that you have set up to assess*

**Reply**:OK, We delete these sections.

*29. L150-L156 – You present the results from Yu et al (2015) and Moore et al (2015), but these will necessarily differ to your results as you use 6 models (you include NorESM-1) and they use 7 models (CSIRO-MK3L and GISS-E2-R). Please state the temperature changes in your ensemble of models, and preferably include ranges. Consider also moving this to the results section of the manuscript*

Yes, we delete this text and use revised numbers in the results section

The climate response to G4 forcing has been discussed by Yu et al. (2015). The general pattern of temperature change under GHG forcing includes accentuated Arctic warming, and least warming in the tropics. G4 largely reverses these changes, but leaves some residual warming in the polar regions and under-cools the tropics. SAI also reduces temperatures over land more than over oceans relative to GHG, and hence reduces the temperature difference between land and oceans. Between 2020 and 2069, SSTs in the 6 basins during their TC seasons are 0.4°C (with a model range of 0.2 to 0.6°C) warmer in RCP4.5 than under G4.

*30. L162 – Sentence beginning 'It is, however,' should have a suitable reference*

**Reply**:Rewritten as It is, however, more interesting for TC studies because the sulphate aerosol injected into the stratosphere causes radiative heating (Pitari et al., 2014), and other indirect effects on the upper troposphere (Visioni et al., 2018)…

*31. L170 – replace 'vector' with 'wind'*

**Reply**:Done.

*32. L172 – define Cp in Equation (2)*

**Reply:** Cp is the heat capacity of dry air at constant pressure

*33. L174 – why do you use the 100 hPa level for the outflow temperature? Do you have a suitable reference?*

**Reply:** We use Wing, A. A., K. Emanuel, and S. Solomon (2015), On the factors affecting trends and variability in tropical cyclone potential intensity, *Geophys. Res. Lett.*, *42*, 8669–8677, doi:10.1002/2015GL066145 and add some sentences in Section 3.3 on this choice. Wing et al. (2015.) use the trends in reanalysis and radiosonde products at 70 and 100 hPa in TC seasons to represent change in outflow temperature across various TC basins and assign its contribution to trends in $V_{pot}$. For convenience we choose the tropical tropopause (100 hPa) temperature from the ESM output to represent $T_O$

*34. L177 – please define what the GPI is, i.e. the theoretical maximum intensity, and what increases/decreases to GPI signify (with a suitable reference)*

**Reply:** We define exactly what GPI is by equation later in the text, and it was discussed during the introduction along with other TC proxies. We are not sure exactly what should be added here. We do rewrite the introductory sentences to the section slightly: The GPI has been widely employed to represent TC activity (e.g., Song et al., 2015), and several different formulations have been described (e.g., Emanuel, 2004; 2010). Here, we chose to use perhaps the most commonly-used method

We assess the large-scale environmental conditions for TC generation primarily using the GPI, but make use of the VI for comparison purposes.

*35. L188 - please define what the VI is and what increases/decreases to VI signify (with a suitable reference)*

**Reply:** As with point #34 we define VI later by equation later in the text, and it was discussed during the introduction along with other TC proxies. We are not sure exactly what should be added here. We add: In contrast with GPI where increases correspond to heightened TCs, increases in VI mean fewer TCs are likely.

*36. L190 – 'greenhouse gas' - > RCP4.5*

**Reply:** Done.

*37. L193 – 'air temperature' on levels or near surface air temperature?*

**Reply:** Air temperature on different vertical levels .

*38. L200 – 'researchers' -> 'studies'*

**Reply:** Done.

*39. L200 – Emanuel 2010 is not a suitable reference as it makes no predictions for the future, only studying observations from 1908-1958*

**Reply**:Actually what we discussing is observations, the reference for Emanuel is correct, but the Knutson reference should be 2010. Rewritten as Some studies (Emanuel, 2010; Knutson et al., 2010) find robust or significant declines in the frequency of events in the Southern Hemisphere, while the Northern Hemisphere is relatively constant in the observational record.

*40. L200 – 'find' -> 'predict'*

**Reply**:Actually what we discussing is observations, see #39 and changed reference to Knutson et al., 2010

*41. L201 – 'in the Southern Hemisphere' … under global warming*

**Reply**:Actually what we discussing is observations, see #39 and changed reference to Knutson et al., 2010, Southern Hemisphere, while the Northern Hemisphere is relatively constant the observational record.

*42. L201 – 'but increasing frequency in the northern hemisphere' – Knutson et al (2015) find no such thing! Sure, they find an increase in the East North Pacific and North Indian basins, but they also predict a decrease in the North Atlantic and West North Pacific basins!*

**Reply**:Actually what we discussing is observations, the Knutson reference should be 2010. Rewritten as Some studies (Emanuel, 2010; Knutson et al., 2010) find robust or significant declines in the frequency of events in the Southern Hemisphere, while the Northern Hemisphere is relatively constant in the observational record.

*43. L203 – 'The observed TC annual-mean numbers for the period 1980-2008 for each basin are also listed in Table 2' – where did these numbers come from? I can only find the basin boundaries in Emanuel 2010. Are these numbers consistent with your basin boundaries? Please provide a suitable reference*

**Reply**:These numbers are from the Figure 3 in Emanuel, (2010). And our basin boundaries are consistent with the basin boundaries in Emanuel, (2010).

*44. L209 – 'annual' -> 'annual-mean'*

**Reply**:Done.

*45. L210 – you use August to October for the Northern Hemisphere, but the North Indian basin has two peaks in activity (one in May) (Li et al., 2013). How does GPI and VI change in this second peak in the Indian basin. Please comment on this.*

**Reply**:We list the model, month and basin GPI and VI in Table S1. Between 2020 and 2069 the GPI under RCP4.5 in May in the North Indian basin is 44, while under G4 it

is 55. BNU-ESM, HadGEM2-ES, MIROC–ESM and NorESM1-M both show clear secondary peaks in GPI in April, May, June, the other models do not. Significant differences (p<0.05 t-test and Wilcoxon signed rank test) are found for MIROC-ESM, MIROC-ESM-CHEM and NorESM1-M. The secondary may peak is much smaller than the general northern hemisphere peak and we redefine our TC season in any case – see point #46. We add: Li et al. (2013) note that the Northern India TC basin has a secondary peak in TC around May. This peak is reproduced by the BNU-ESM, HadGEM2-ES, MIROC–ESM and NorESM1-M models where it about half the size of the peak months later in the year (Table S1). This does not affect the statistical choice of TC months (Table S2), although it causes the fraction of GPI accounted for in our TC season to be the lowest for the Northern Indian basin (Table S3).

*46. L210 – what percentage of total annual storms in each basin occur during your chosen timeframes? These 3 month timeframes seem very narrow to me*

**Reply:** After reading your points and the major point #5 we reassessed our choices. We list the basin GPI and VI by model and month in Table S1.The individual monthly GPI as a fraction of the annual totals are shown in Table S2. We select northern and southern TC season on the basis of the each model's monthly fractions of GPI. We use a threshold of 10% for above uniformly distributed GPI for RCP4.5 and G4 averaged GPI and find that for the northern basins June-November are above the threshold, while for the southern basins it is January-June. Thus there are 6 months in each hemisphere and they account for 68% under RCP4.5 and 69% under G4 of the yearly total GPI (Table S3). We also notice from Table S2 that under G4 the TC season occurs about 1 month earlier than under RCP4.5 in both hemispheres, although our choice of threshold for the TC season means that we can use the same 6 months for each experiment. While peak TC season. The same analysis for VI shows similar results, although the season is less well-defined than for GPI, for instance VI in August is higher than December in northern basins as is January in the southern ones, but the general results do not require separate definitions of season from those for GPI. The Northern Hemisphere peak TC season is August through October and January through March in the Southern Hemisphere season, various authors have used longer periods in analyzing model data, e.g. Emanuel (2013) used all 12 months, while Jones et al., (2017) used June-November for the North Atlantic hurricane season. Li et al. (2013) note that the Northern Indian TC basin has a secondary peak in TC around May. This peak is reproduced by the BNU-ESM, HadGEM2-ES, MIROC–ESM and NorESM1-M models where it about half the size of the peak months later in the year (Table S1). This does not affect the statistical choice of TC months (Table S2), although it causes the fraction of GPI accounted for in our TC season to be the lowest for the Northern Indian basin (Table S3).

*47. L217 – 'Furthermore, the G4 means for all models were significantly lower than*

*their RCP4.5 values' – Table 3 seems to say the opposite, that none of the changes to GPI are significant! Please give annual-mean values, standard deviations and p-values for G4 and RCP4.5, perhaps in Table 1?*

**Reply**:There was some confusion what trends were significant, and what was different under G4 and RCP4.5 that we have now resolved. The revised Table 3, new Table 4 and Fig. 1 taking account of the new TC seasons (#46) shows clearer differences. We also produce 3 new tables in the SI that show monthly and basin model results, we prefer this solution than adding results to our Table 1 as the referee suggests. We write: The models we use have considerable range in their absolute values of GPI, which is also a generally observed feature of climate models (Emanuel, 2013). The GPI has a rising trend under RCP4.5 and G4 (Fig. 1). Table 3 shows that there are significantly ($p<0.05$ when tested using the Wilcoxon signed rank test) lower values of GPI under G4 than RCP4.5 for Northern Hemisphere basins in all models, but only MIROC-ESM-CHEM has significantly lower GPI for the Southern Hemisphere basins. The time series indicate that tropical storms will become more frequent with time and that G4 significantly reduces the numbers.

*48. L221 – 'The time series indicate that tropical storms will become more frequent with time and that G4 significantly reduces numbers' – Fig. 1 does not indicate this at all to me! There are many years, for each model, where the GPI is higher for G4 than for RCP4.5. Figure 1 will need rethinking as it does not support the central tenet of your paper. How for instance, can a difference of -0.3 % in CanESM2 be significant?*

**Reply**:See reply to #47. Revised TC season numbers in Table 3: Table 3 shows that there are significantly ($p<0.05$ when tested using the Wilcoxon signed rank test) lower values of GPI under G4 than RCP4.5 for Northern Hemisphere basins in all models except for NorESM1-M, but only MIROC-ESM-CHEM has significantly lower GPI for the Southern Hemisphere basins.

*49. L226 – Sentence starting 'During most years from 2020 to 2069…' – this is hardly a sufficient statistical test for significance, simply saying VI looks higher for G4 than RCP4.5! Please perform significance tests and identify which models show significant VI changes and which ones don't. Table 3 suggests that no VI changes are significant!*

**Reply**:Yes, we explicitly now state that we test significance using both Student's t-test (Table S1) and Wilcoxon signed rank test (Tables S1 and 3). We rewrite: Fig. 1 also shows the evolution of VI in the TC seasons during 2020 to 2069 among the five models. Note that following the definition of VI in Tang et al. (2014) we use the median value not its mean. All models show decreasing trends over time, indicating a tendency for more TCs, consistent with trends in GPI. Table 3 shows that G4-RCP4.5 differences in Northern Hemisphere basins are significantly positive except for NorESM1-M, Southern Hemisphere basins show less consistent results, which is also consistent with GPI which indicates that G4 reduces TC occurrence, and is more effective in the

*Table 3. Differences (G4-RCP4.5) in TC basins and season during 2020-2069 year calculated point-by-point. Northern Hemisphere numbers are above and Southern Hemisphere below Bold fonts are significant at 95% level using the Wilcoxon signed rank test. The ensemble means are not normalized.*

| Models | $T_s$ (°C) | $T_o$ (°C) | $T_s$-$T_o$ (°C) | GPI (%) | $V_{pot}$ (ms⁻¹) | $H$ (%) | $V_{shear}$ (ms⁻¹) | $\eta$ (×10⁻⁸ s⁻¹) | VI (%) | $\chi_m$ (×10⁻³) |
|---|---|---|---|---|---|---|---|---|---|---|
| BNU-ESM | **-0.50** | **0.12** | **-0.62** | **-3.8** | **-0.45** | -0.071 | 0.014 | -0.63 | **2.2** | **16** |
|  | **-0.42** | 0.11 | **-0.53** | 0.37 | 0.070 | 0.20 | **-0.27** | -1.0 | **-1.5** | **15** |
| MIROC-ESM | **-0.34** | **-0.58** | **0.24** | **-6.7** | **-0.94** | **-0.36** | 0.13 | 1.3 | **2.5** | -3.7 |
|  | **-0.30** | **-0.56** | **0.26** | -0.86 | **-0.50** | -0.19 | 0.13 | -2.3 | **2.3** | 6.8 |
| MIROC-ESM-CHEM | **-0.25** | **-0.45** | **0.21** | **-4.8** | **6.9** | **4.8** | **1.8** | -0.054 | **1.9** | **-7.9** |
|  | **-0.21** | **-0.43** | **0.22** | **-11** | **6.5** | **3.6** | **2.2** | -0.027 | **1.3** | 3.6 |
| NorESM1-M | **-0.23** | **-0.087** | **-0.15** | **4.8** | **-0.52** | **-0.51** | 0.029 | **-3.4** | **-2.0** | -4.8 |
|  | **-0.21** | -0.071 | **-0.14** | -0.73 | **-0.62** | -0.10 | -0.12 | -0.83 | **2.5** | 3.3 |
| HadGEM2-ES | **-0.65** | **0.16** | **-0.80** | **-3.1** | **-1.0** | 0.17 | 0.041 | **1.9** | **3.8** | 35 |
|  | **-0.61** | **0.15** | **-0.76** | 0.39 | **-0.71** | -0.088 | -0.079 | 1.0 | **1.1** | 30 |
| Ensemble | **-0.40** | **-0.14** | **-0.26** | **-2.7** | **0.80** | **0.80** | **0.40** | -0.2 | **1.9** | 7.0 |
|  | **-0.35** | **-0.13** | **-0.23** | **-2.5** | **0.95** | **0.68** | **0.37** | -0.7 | **1.0** | 11.8 |

*50. L231 – 'As with GPI, there is about a factor of 2-3 range in absolute values between the models' – perhaps plot normalised anomalies relative to 2020-2030 in Figure 1 instead?*

**Reply**:We experimented with many different ways of plotting, and now use the methods shown, with a different method for VI than GPI because of the separation of the model results.

[Figure]

**Figure 1.** Five yearly moving annual averages across the 6 TC basins and TC season, of (a) GPI, solid lines denote forcing under RCP4.5 and dotted lines values under G4. Ensemble mean series were calculate using normalized time series, shifted by the ensemble mean. (b) VI with solid lines denoting model ensemble means and shading indicating the range across the five models.

Since we recalculate with new TC season all figures in the paper have been revised, and their associated text. We rewrite: Fig. 2 shows the correlations between model differences G4-RCP4.5 for annual mean GPI and VI. Most models, and the ensemble show significant anti-correlation across all TC basins, except the South Pacific where half the models have significant correlation. The ensemble mean correlation is only around -0.3, indicating that GPI and VI are addressing sufficiently different aspects of TC to warrant independent analysis.

*51. L245 – 'All models except NorESM1-M show negative differences in the North*

*Indian basin' – this may be true on a basin-wide basis, but BNU-ESM shows GPI increases in the Bay of Bengal. This might indicate a change in the spatial distribution of storms in the North Indian basin. This is also apparent in the ensemble-mean*

**Reply**:Actually with the new TC season BNU-ESM has no clear sign of response in NI basin, so our statement is better than the referee's suggestion.

*52. L269 – consider changing 'item' to 'component' or 'term' throughout this paragraph*

**Reply**:Done.

*53. L274 – has this decomposition (Eq. 5) been used before? If so, provide a relevant reference*

**Reply**:Yes, it is from Li et al., 2013. We rewrite as : Li et al. (2013) expressed Equation (1) for GPI as the product of four terms, respectively representing an atmospheric absolute vorticity term ($AV$), a vertical wind shear term ($WS$), a relative humidity term ($RH$), and an atmospheric potential intensity term ($PI$).

*54. L301 – 'Hence, these are the factors that primarily enable solar geoengineering'*

**Reply**:Done.

*55. L304 – do you have any idea as to why MIROC-ESM-CHEM is so different?*

**Reply**:Yes. Firstly there was error in analyzing the MIROC-ESM-CHEM ensemble. Second, the 9 members divide into 2 separate groups in terms of how much variance the explanatory variables make to linearized GPI and VI. On this basis we exclude CanESM2 from the analysis completely, and add new supplementary figures to show how MIROC-ESM-CHEM members differ from each other. In section 2 we write Although to date 8 ESM have performed the RCP4.5 and G4 simulations, a subset of 6 models have access to all required model data fields, but one of those, CanESM2, was not used because all three of the realizations available it failed to pass statistical tests leaving 5 models (Table 1). The particular tests we did to exclude some data and models from the analysis are discussed in detail in section 3.2. The rejected simulations all produced statistically weak and insignificant regression fits to linearized forms of GPI and VI with all combinations of the thermodynamic and dynamic terms used to compute them. Hence, it is unlikely that VI or GPI can meaningfully represent TC activity in these cases. In comparison, the ESM simulations we do use have regression models that are significant at least at the 5% level, and in many cases, achieve far higher significance.

In section 3.2.2 Fig. S1 shows the same analysis as Fig. 5, but for all 9 realizations of MIROC-ESM-CHEM. The first four realizations behave similarly as the BNU-ESM, HadGEM2-ES and MIROC-ESM models in Fig. 5, with variance accounted for around

80% of total and the *RH* terms being about twice as important as *WS* and *PI* terms. The remaining 5 realizations have far lower variance explained, similar as for NorESM1-M, with *RH* still the dominant term…. Fig. S3 shows the VI components for all 9 realizations of MIROC-ESM-CHEM, which appears similarly divided into two groups as they were for GPI in Fig. S1…. When we analyzed the realizations 5-9 of MIROC-ESM-CHEM we found much lower F-statistics than for realizations 1-4 (Fig. S4), with values similar as for NorESM1-M of 50-100. In general, the models show *RH* has the largest F-statistic for single parameter models, consistent with Figs. 4 and 5. Fig. S4 also shows that all three realizations of CanESM2, which we do not use for TC analysis in this paper, have even lower F values, particularly r2 and r3, which are around 2.

[Figure]

**Figure S1**. As Fig. 5 but for the 9 realizations of MIROC-ESM-CHEM: The fractional variance contribution of components of GPI during the TC season and within the six TC basins during 2020-2069.

[Figure]

**Figure S3**. As S2 but for the 9 realizations of MIROC-ESM-CHEM: The fractional variance contribution of components of VI during the TC season and within the six TC basins during 2020-2069.

[Figure]

**Fig. S4** As Fig6: The F-statistic of the 15 different combinations of regression variables for GPI differences between G4 and RCP4.5, but for each realizations 1-9 of MIROC-ESM-CHEM, (top 3 rows, and for the 3 realizations of CanESM2 (bottom row). The x-axis on each panel represents the combination of components used as predictors in each regression equation: 1:*(PI,RH,WS,AV)*, 2:*(PI,RH,WS)*, 3:*(PI,RH,AV)*,   4:*(AV,RH,WS)*, 5:*(PI,AV,WS)*, 6:*(PI,RH)*, 7:*(PI,WS)*, 8:*(PI,AV)*, 9:*(RH,WS)*, 10:*(RH,AV)*, 11:*(AV,WS)*, 12:*(PI)*, 13:*(RH)*, 14:*(WS)*, 15:*(AV)*.

*56. L312 – Sentence starting 'Fig. 4 shows that the HadGEM2 values tend to be smaller' – CanESM2 is similarly muted and seems to have the same signs as HadGEM2*

**Reply:** In fact the new Fig. 3 and 4 shows that HadGEM2 results are not strikingly muted or different form the other models. So we delete that part of the sentence.

*57. L356 – 'The key factors affecting TCs' – consider adding a more informative title, possibly, 'Primary factors that control GPI and VI changes'?*

**Reply:** Changed as suggested.

*58. L357 – You seem to have found in Section 3.2 that relative humidity is the most important factor for GPI, and then ignore this finding in Section 3.3, which I found curious*

**Reply:** Actually we discuss *RH* later but we move that paragraph up and introduce the main ideas before discussing *PI*.

The analysis above shows that the common factors across models and basins that affect TCs are potential intensity ($V_{pot}$), relative humidity ($H$), and vertical wind shear ($V_{shear}$).

We now discuss these factors separately, beginning with $V_{pot}$ as this is function of several different ESM variables.

*59. L368 – 'The model ensemble' -> 'The ensemble-mean'*

**Reply:** Done.

*60. L370 – 'Fig S3 shows that correlations for both models under RCP4.5 and G4 separately are not atypical, simply that their G4-RCP4.5 differences are small' – I'm not sure what you mean by this sentence, please rephrase*

**Reply:** With the new TC seasons this sentence is no longer needed and deleted. Fig. 7a shows the dependence of $V_{pot}$ differences (G4-RCP4.5) on ($T_S - T_O$) differences for the models. All models have significant correlation for all TC basins except BNU-ESM in the SI and SP basins and HadGEM2-ES in the SP basin. However, there is an even stronger dependence for $V_{pot}$ on $T_S$ anomalies (Figs. 7b, S3). The ensemble mean $V_{pot}$ is better correlated with $T_s$ rather than ($T_S - T_O$) due to better correlations of all models in all basins except HadGEM2-ES.

*61. L373 – Similarly the last sentence is unclear. Do you mean that all models excepts*

*CanEMS2 and NorESM1 exhibit significant correlation between Ts and Vpot in all basins?*

**Reply :** This sentence is deleted and replace with All models show significant correlation between GPI and $T_S$ anomalies shown as Fig. 7c. Some models have insignificant correlations in particular basins, e.g., BNU-ESM is slightly anti-correlated in NA, as is HadGEM2-ES in WNP. GPI is not significantly correlated with $T_s$ for half the ESM in the NI and SP basins. Fig. S3 shows that there are fewer significant correlations under G4 than under RCP4.5.

*62. L376 – Change 'variability' to 'cycle' throughout this paragraph*

**Reply:** Done.

*63. L384 – Remove 'Comparing'*

**Reply:** Done.

*64. L388 – The last sentence – can you also plot the seasonal cycle of Ts in ERA-interim just to confirm that all the models are doing reasonably well here?*

**Reply:** Added to renumbered Fig. S9.

*65. L390 – Consider splitting this paragraph into two. It is too long and unwieldy as it is*

**Reply:** Done.

*66. L391 – Sentence beginning 'In Figs 7d and 7e we plot' – please reword this sentence to something like 'we plot correlations between H / Vshear and Ts.*

**Reply:** We plot $H$ differences between G4 and RCP4.5 as a function of sea surface temperature differences in Fig. 7d.

*67. L399 – 'there is generally an anti-correlation between Vshear and    Ts'*

**Reply:** rewritten as: Fig 7e shows how RCP4.5-G4 differences in $V_{shear}$ and $T_s$ are generally anti-correlated. The across-model spread for correlations of $V_{shear}$ and $T_s$ under both G4 and RCP4.5 (Fig. S3) are similar as for the other key variables. Anti-correlation with $T_s$ is weakest in the SP and NA basins, but still significant. In terms of the differences in Fig. 7e, all models show clear significant anti-correlations, with the NI and NA basins having weakest correlations.

*68. L402 – Vecchi and Soden (2007) found that wind shear increases in both the North Atlantic and the East Pacific under global warming.*

**Reply:** Rewritten as Vecchi and Soden (2007) found the North Atlantic and East North

Pacific wind shear increases in model projections under global warming. If the models assessed here capture the effect under G4 and RCP45, we would expect positive correlations between $V_{shear}$ and $T_s$ over these two basins for G4 and RCP4.5 in Fig. S3.

*69. L404 – 'If the models assessed here'*

**Reply**:Done.

*70. L407 to L415 – I'm not sure what you are trying to prove here, it seems peripheral and needs to be reworded*

**Reply**:We delete this section and Fig. S7, though we use the reference in the discussion where it may make our point clearer than it was: The final variable, $V_{shear}$ , shows large scatter across the models, but consistent anti-correlation with $T_s$. However, there are also good but different relations between $H$ and $V_{shear}$ in every basin suggesting that the state of this dynamic variable can be explained to a significant degree by the thermodynamic state driving $H$ and $T_s$. This is consistent with analysis (Li et al., 2010), showing that prescribed sea surface temperatures can account for some changes in TC in the Pacific basins as surface temperature gradients drive trade winds, which changes the wind shear

We deleted sections 3.4 and 3.5 so points #71 - 75 are moot

*71. L441 – 'The analysis for individual basins indicates most models have significant correlations with ENSO in the WNP' – This is not true! Only 4/7 of the models have significant correlation in the WNP in RCP4.5*

*72. L448 – is there any previous studies that suggest a link between ENSO and tropical cyclone activity in basins outside the North Atlantic and the Pacific. If so, please cite*

*73. L449 – 'is most consistently felt in the Pacific Ocean'–particularly the South Pacific*

*74. L473 – why are the TRACK results so much lower in your Table 4 than in Jones et al. (2017)? For instance, you get 1.2 storms per year in the North Atlantic basin in G4 compared to ~11 per year in their work (their Fig. 4). Their reasoning behind the use of the (4.5,3.5,4) configuration was to attain ~10 storms per year on average in the historical period. Please check these numbers, they seem wrong.*

*75. L487 – Change 'typical' to 'current'*

We delete sections 3.4 and 3.5 so points #71 - 75 are moot

*76. L489 – 'The storms that may be counted using indirect methods such as the TRACK algorithm include the whole climate condition' – This doesn't make sense to me. Consider replacing with 'Simulated storms that may be counted using methods such as the TRACK algorithm allow for feedbacks with the climate system'*

**Reply**:Simulated storms that may be counted using methods such as the TRACK

algorithm (Hodges, 1995; Jones et al. 2017) that allow for feedbacks with the climate system.

*77. L490 – 'Statistical methods (Moore et al., 2015) also implicitly include feedbacks between storm and climate conditions' – in what way do they include feedbacks? I don't understand this. They are simply diagnostics*

**Reply:** They do include feedbacks in their maps of teleconnections because they consider the non-local changes in e.g. the surface temperature that are both caused by, and that cause Atlantic hurricanes. Thus cooling over the USA related to extreme hurricanes is because of the cooling they produce, while heating over deserts is related to factors that lead to Atlantic hurricanes. We expand slightly the sentence: Statistical methods (Moore et al., 2015) may also implicitly include feedbacks between regional storm and background global climate conditions

*78. L492 – 'but dynamical downscaling methods (Emanuel, 2013) cannot include them' – I disagree, Emanuel employs a simple ocean model which can be adjusted to provide climate feedback. In fact, I think the semi-explicit scheme offers more opportunity to incorporate feedbacks than the statistical methods*

**Reply:** But that means it has to be manually adjusted rather than automatically occurring because of TC events. In that sense it does not include a feedback as the statistical methods do. We slightly rewrite to say but dynamical downscaling methods (Emanuel, 2013) do not include them

*79. L493 – change 'apply' to 'utilize'*

**Reply:** Done.

*80. L495 – change 'relatively little data' to 'coarse temporal-resolution data'*

**Reply:** Done.

*81. L502 – Change 'diagnose tropical storms in climate models' to 'relate tropical storm activity to ambient meteorology'*

**Reply:** Done.

*82. L507 – 'Thus stratospheric sulphate aerosol injection could lead to fewer TCs in the North Atlantic …' – note that this is one solar geoengineering scenario (a uniform one). Injecting aerosol preferentially into one hemisphere may increase the amount of storms in the North Atlantic (Jones et al (2017)) with unknown effects in other basins*

**Reply:** Rewritten as: Thus the G4 scenario of SAI based on equatorial lower stratosphere injection of $SO_2$ could lead to fewer TCs in the North Atlantic and Indian Ocean but more TCs in the South Pacific region than under GHG induced global warming

*83. L510 – 'The impact of ENSO on TCs can be detected in the GPI' – this is poorly worded, you have not explicitly looked at ENSO and TCs, only at ENSO and GPI. Rephrase in such a way: 'ENSO is found to be correlated with GPI'. Are there any implications specifically in terms of solar geoengineering from your results? I mean, is there a decrease in El Nino years in the G4 simulations?*

**Reply**:We delete discussion of ENSO and this sentence.

*84. L515 – remove 'such as'*

**Reply**:Done.

*85. L521 – 'a simplified representation of TCs depending on fewer variables is possible' > 'a simplified representation of the GPI depending on fewer variables may be possible'*

**Reply**:Done.

*86. L523 – sentence running from 'it is encouraging that the thermodynamic state …' I don't understand what you mean here?*

**Reply**:Rewritten to clarify that local factors are also important: Although wind shear is important and a dynamic variable, it in encouraging that the thermodynamic state of the system is of prime importance for the GPI. This suggests that statistical methods of predicting changes in TC behavior are plausible, although individual basin behavior depends on particular local forcing factors in addition the accessible thermodynamic variables used in the GPI and VI.

*87. L529 – '(the 100hPa level)' -> (evaluated at 100 hPa)'*

**Reply**:Done.

*88. L529 – Replace 'note that' with 'find that changes to' and add 'changes' after GPI*

**Reply**:Done.

*89. L542 – rather than using temperature changes from Pitari et al (2014), can you give the ensemble mean upper-tropospheric temperature changes from your 6-member ensemble please*

**Reply**:We do give this in Table 3, as we say in the text, assuming that 100 hPa temperatures represent the upper troposephere values. Pitari et al., (2014) note a warming of the 100 hPa layer under G4 relative to RCP4.5 for the MIROC-ESM-CHEM model in the 2040s for the tropics. Most models (Table 3) in the TC basins and seasons show a cooling of (ensemble mean of 0.14°C) with only HadGEM2-ES and BNU-ESM having warming at 100 hPa. Given the complexities of changes in the upper troposphere due to the process outlined in the previous paragraph the range in static stabilities represented by the model range in $T_s$-$T_o$ differences relative to RCP4.5 is

probably not surprising. Therefore, although we might expect to see an improvement in correlation of potential intensity and GPI by using 100 hPa temperatures in addition to SSTs, the ability of the models to capture all the processes varies. The result is that the models used here have a better relationship with sea surface temperatures than static stability, and suggests that the aerosol effects are not being properly simulated to allow their impacts on TC genesis to be fully estimated.

*90. L542 – 'This is about half the range of the G4-RCP4.5 difference in static stability (Fig. 7)' – Figure 7 does not show that changes to static stability…*

**Reply**:We rewrite this section:In contrast with the solar dimming G1 experiments analyzed by Davis et al., (2016), here we analyse G4 which is an aerosol injection protocol. The aerosol is prescribed in the GeoMIP G4 protocol (Kravitz et al., 2011a) as injected into the equatorial stratosphere at 16-25 km altitude, where most of the direct radiative heating takes place (Pitari et al., 2014). However, due to the large size of the geoengineering aerosol particles (effective radius of the order of 0.6 μm or more), a significant fraction of the stratospheric particles settle below the tropical tropopause (Niemeier et al., 2011; English et al., 2012; Cirisan et al, 2013), thus producing some diabatic heating a few kilometres immediately below the tropical tropopause. This is superimposed on the convectively-driven upper tropospheric cooling caused by surface cooling due to the SAI and reduced convection and weakened hydrological cycle (Bala et al., 2008). This may be expected to be the dominant process controlling the SAI-induced changes in atmospheric static stability. Furthermore, recent work (Visioni et al., 2018 ACP in discussion) explores the secondary of surface cooling on the upper troposphere with the impact on cirrus clouds, and the concomitant impact on static stability. Surface cooling and lower stratospheric warming, together, tend to stabilize the atmosphere, thus decreasing turbulence and water vapor updraft velocities. The net effect is an induced cirrus thinning, which serves to increase net global cooling due to the SAI.

*91. L544 – remove 'significant'*

**Reply**:Done.

*92. L545 – why does T0 not warm with most models under G4? Do you have a reason that you can offer? It is that the aerosol particles are small?*

We rewrite this part to try to answer these questions using suggestions from Ref #2. In contrast with the solar dimming G1 experiments analyzed by Davis et al., (2016), here we analyse G4 which is an aerosol injection protocol. The aerosol is prescribed in the GeoMIP G4 protocol (Kravitz et al., 2011a) as injected into the equatorial stratosphere at 16-25 km altitude, where most of the direct radiative heating takes place (Pitari et al., 2014). However, due to the large size of the geoengineering aerosol particles (effective radius of the order of 0.6 μm or more), a significant fraction of the stratospheric

particles settle below the tropical tropopause (Niemeier et al., 2010; English et al., 2012; Cirisan et al, 2013), thus producing some diabatic heating a few kilometres immediately below the tropical tropopause. This is superimposed on the convectively-driven upper tropospheric cooling caused by surface cooling due to the SAI and reduced convection and weakened hydrological cycle (Bala et al., 2008). This may be expected to be the dominant process controlling the SAI-induced changes in atmospheric static stability. Furthermore, recent work (Visioni et al., 2018 ACP in discussion) explores the secondary of surface cooling on the upper troposphere with the impact on cirrus clouds, and the concomitant impact on static stability. Surface cooling and lower stratospheric warming, together, tend to stabilize the atmosphere, thus decreasing turbulence and water vapor updraft velocities. The net effect is an induced cirrus thinning, which serves to increase net global cooling due to the SAI.

Pitari et al. (2014) note a warming of the 100 hPa layer under G4 relative to RCP4.5 for the MIROC-ESM-CHEM model in the 2040s for the tropics. Most models (Table 3) in the TC basins and seasons show a cooling of (ensemble mean of 0.14°C) with only HadGEM2-ES and BNU-ESM having warming at 100 hPa. Given the complexities of changes in the upper troposphere due to the process outlined in the previous paragraph the range in static stabilities represented by the model range in $T_s$-$T_o$ differences relative to RCP4.5 is probably not surprising. Therefore, although we might expect to see an improvement in correlation of potential intensity and GPI by using 100 hPa temperatures in addition to SSTs, the ability of the models to capture all the processes varies. The result is that the models used here have a better relationship with sea surface temperatures than static stability, and suggests that the aerosol effects are not being properly simulated to allow their impacts on TC genesis to be fully estimated.

*93. L578 – 'Many models, owing to their low resolutions, produce much weaker and larger TC' – this statement has been repeated a few times (e.g. L487). Please do not repeat statements*

**Reply:** Rewritten as: Considering the coarse spatio-temporal resolution of most ESM models, evaluating the GPI is likely to remain a popular be a good diagnostic of TC variations under different climates.

*94. L582 – change 'would be' to 'this is'*

**Reply:** Done.

---

## Author Comment (AC2) · 21 May 2018

In the reply, the referee's comments are in *italics*, our response is in normal text, and quotes from the manuscript are in blue.

***Anonymous Referee #2***

*In this numerical work, a statistical approach is described for analysing the effects of sulphate geoengineering on the genesis of tropical storms. The procedure is well designed on the general methodology of the GeoMIP project, with use of data that independent global models have provided in a common database with their G4 simulations. The manuscript is scientifically robust and deserves publication on ACP.*

*Some of the conclusions are important, mainly the fact that the thermodynamic role of SST changes induced by geoengineering aerosols dominates over the lower stratospheric aerosol heating. However, sometimes the authors compare the SST effects with changes in static stability, as if they were two independent things (see for example in the conclusions, lines 547-549). Actually, SST changes may affect the atmospheric static stability by themselves, even in the absence of a stratospheric warming. I would suggest rephrasing. The authors themselves clearly explain how static stability changes are controlled by both surface and upper tropospheric temperatures (page 18, lines 358-360). This is the main specific point I suggest to better clarify all along the manuscript, before final publication on ACP.*

**Reply**:Yes, this is good point. We fully appreciate the point that static stability is not the same as SST. Apparently our original sentences were not clear enough on this and we have rewritten the entire section discussing impacts on static stability due to SAI, with the helpful suggestions from the referee.

*In addition, it is true that the aerosol heating is mostly located in the 16-25 km layer (see page 27, lines 540-542); however, due to the large size of the geoengineering aerosol particles (effective radius of the order of 0.6 μm or more), a significant fraction of the stratospheric particles would settle down below the tropical tropopause (Niemeier et al., 2010; English et al., 2012; Cirisan et al, 2013), thus producing some diabatic heating superimposed to the convectively-driven upper tropospheric cooling. This means that the surface cooling (with associated upper tropospheric tropical cooling, due to lesser efficient convective motions) may be expected as the dominant process controlling the geoengineering induced changes of atmospheric static stability. At the same time, the aerosol heating in a few kilometres layer immediately below the tropical tropopause (due to gravitational sedimentation of large geoengineering sulfate aerosols) should also be considered as a contributing smaller effect.*

**Reply**:Thank you for this insight. We modify the text to take these points into account: In contrast with the solar dimming G1 experiments analyzed by Davis et al., (2016), here we analyze G4 which is an aerosol injection protocol. The aerosol is prescribed in the GeoMIP G4 protocol (Kravitz et al., 2011a) as injected into the equatorial stratosphere at 16-25 km altitude, where most of the direct radiative heating takes place (Pitari et al., 2014). However, due to the large size of the geoengineering aerosol particles (effective radius of the order of 0.6 μm or more), a significant fraction of the stratospheric particles settle below the tropical tropopause (Niemeier et al., 2010; English et al., 2012; Cirisan et al, 2013), thus producing some diabatic heating a few kilometres immediately below the tropical tropopause. This is superimposed on the convectively-driven upper tropospheric cooling caused by surface cooling due to the SAI and reduced convection and weakened hydrological cycle (Bala et al., 2008). This may be expected to be the dominant process controlling the SAI-induced changes in atmospheric static stability

*It would be worth to note that another indirect effect of sulfate geoengineering, related to the surface cooling and static stability changes, is discussed in Visioni et al. (2018). Here the sensitivity of upper tropospheric ice formation is studied with inclusion of the aerosol-induced surface cooling, with respect to a reference condition documented in Kuebbeler et al. (2016), where only the stratospheric warming due to the aerosols was taken into account. The conclusions presented in the manuscript of Wang et al. (2018) go in the same direction of what discussed in this other study.*

**Reply**:Yes, thank you we not this now: Furthermore, recent work (Visioni et al., 2018 ACP in discussion) explores the secondary of surface cooling on the upper troposphere with the impact on cirrus clouds, and the concomitant impact on static stability. Surface cooling and lower stratospheric warming, together, tend to stabilize the atmosphere, thus decreasing turbulence and water vapor updraft velocities. The net effect is an induced cirrus thinning, which serves to increase net global cooling due to the SAI.

*Minor points*

*P. 3, line 66: the Kravitz reference has a wrong comma between the name and et al.*

**Reply**:Done

*P. 3, line 72: some more recent articles can be cited here, for example Visioni et al. (2017).*

**Reply**:yes we added Visioni (2017); Kashimura, H., M. Abe, S. Watanabe, T. Sekiya, D. Ji, J. C. Moore, J.N.S. Cole and B. Kravitz 2017 Shortwave radiative forcing, rapid adjustment, and feedback to the surface by sulfate geoengineering: analysis of the Geoengineering Model Intercomparison Project G4 scenario, *Atmospheric Chemistry and Physics* 17, 3339-3356, doi:10.5194/acp-17-3339-2017, 2017; and Russotto, R. D. and Ackerman, T. P.: Energy transport, polar amplification, and ITCZ shifts in the GeoMIP G1 ensemble, Atmospheric Chemistry and Physics, 18, 2287–2305, doi:10.5194/acp-18-2287-2018, 2018)

*P. 4, line 83: are used instead of have used.*

**Reply**:Done

*P. 6, line 126-130: I would suggest rephrasing this concept, maybe splitting the long sentence in two. In its present form it is hard to follow.*

**Reply**:Rewritten: Jones et al. (2017) showed SAI in the northern hemisphere reduced the numbers of TC in the North Atlantic while SAI in the southern hemisphere increased numbers in the basin.

*P. 7, line 149: explain better the altitude at which the injection is simulated, since it has been shown how different injection heights may affect differently the climate response (Tilmes et al., 2017; Kleinschmitt et al., 2018).*

**Reply**:Rewritten . G4 is based on the GHG emissions from the RCP4.5 scenario but short wave radiative forcing is reduced by injection of $SO_2$ into the equatorial lower stratosphere at altitudes of 16–25 km, at a rate of 5 Tg per year from the year 2020 to 2069.

*P. 8, line 162-165: for a recent study analysing the connection between the stratospheric warming due to the sulfur injection and the tropospheric response in term of vertical motions, see Visioni et al. (2018) (now under review in ACPD).*

**Reply**:  Rewritten, reference added

*P. 14-15: I suggest to the authors to move some of the longer equations derivations to the supplementary material for better readability of the manuscript.*

**Reply**:  Several equations are removed and so improve readability.

*P. 22, line 433: no comma between models and are.*

**Reply**:  This section has been deleted in response to Ref#1.

*P. 25, line 510: I would suggest using "variability" instead of "variations".*

**Reply**:  Done

*P. 27, line 539: analyze instead of analysis. Better rephrase "aerosol injection scheme" into a more appropriate description, such as "protocol".*

**Reply**:  Done.

*P. 27, line 552: "in the response strength across the ocean basins" sounds probably better than "in strength of response across the ocean basins".*

**Reply**:  Done.

*P.38, Fig. 1: adding a legend outside the figure, instead of having the names of the models close to the related lines, would make it easier to read.*

**Reply**:  We have revised Fig. 1 to separate the lines better which we hope solves this problem.

[Figure]

[Figure]

**Figure 1.** Five yearly moving annual averages across the 6 TC basins and TC season, of (a) GPI, solid lines denote forcing under RCP4.5 and dotted lines values under G4. Ensemble mean series were calculate using normalized time series, shifted by the ensemble mean. (b) VI with solid lines denoting model ensemble means and shading indicating the range across the five models.

---

## Short Comment (SC1) · 22 May 2018

updated manuscript can only be uploaded after discussion is complete. Discussion should end today, 22 May. ACP will then send us links to upload a revised manuscript.

---

## Short Comment (SC2) · 22 May 2018

Furthermore, recent work (Visioni et al., 2018 ACP in discussion) explores the surface cooling impact on upper tropospheric cirrus cloud formation, and the concomitant impact on static stability. Surface cooling and lower stratospheric warming, together, tend to stabilize the atmosphere, thus decreasing turbulence and updraft velocities. The net effect is an induced cirrus thinning, which indirectly increases net global cooling due to the SAI.